# Characterization and Learning of Causal Graphs with Latent Variables from Soft Interventions

**Murat Kocaoglu**[*]
MIT-IBM Watson AI Lab
IBM Research MA, USA
murat@ibm.com

**Amin Jaber**[*]
Department of Computer Science
Purdue University, USA
jaber0@purdue.edu

**Karthikeyan Shanmugam**[*]
MIT-IBM Watson AI Lab
IBM Research NY, USA
karthikeyan.shanmugam2@ibm.com

**Elias Bareinboim**
Department of Computer Science
Columbia University, USA
eb@cs.columbia.edu

## Abstract

The challenge of learning the causal structure underlying a certain phenomenon is undertaken by connecting the set of conditional independences (CIs) readable from the observational data, on the one side, with the set of corresponding constraints implied over the graphical structure, on the other, which are tied through a graphical criterion known as d-separation (Pearl, 1988). In this paper, we investigate the more general setting where multiple observational and experimental distributions are available. We start with the simple observation that the invariances given by CIs/d-separation are just one special type of a broader set of constraints, which follow from the careful comparison of the different distributions available. Remarkably, these new constraints are intrinsically connected with do-calculus (Pearl, 1995) in the context of soft-interventions. We then introduce a novel notion of interventional equivalence class of causal graphs with latent variables based on these invariances, which associates each graphical structure with a set of interventional distributions that respect the do-calculus rules. Given a collection of distributions, two causal graphs are called interventionally equivalent if they are associated with the same family of interventional distributions, where the elements of the family are indistinguishable using the invariances obtained from a direct application of the calculus rules. We introduce a graphical representation that can be used to determine if two causal graphs are interventionally equivalent. We provide a formal graphical characterization of this equivalence. Finally, we extend the FCI algorithm, which was originally designed to operate based on CIs, to combine observational and interventional datasets, including new orientation rules particular to this setting.

## 1 Introduction

Explaining a complex system through their cause and effect relations is one of the fundamental challenges in science. Data is collected and experiments are performed with the intent of understanding how a certain phenomenon comes about, or how the underlying system works, which could be social, biological, artificial, among others. The study of causal relations can be seen through the lens of learning and inference [16, 21]. The learning component is concerned with discovering the causal structure, which is the very subject of interest in many domains, since they can provide insight about

---

[*]Equal contribution.

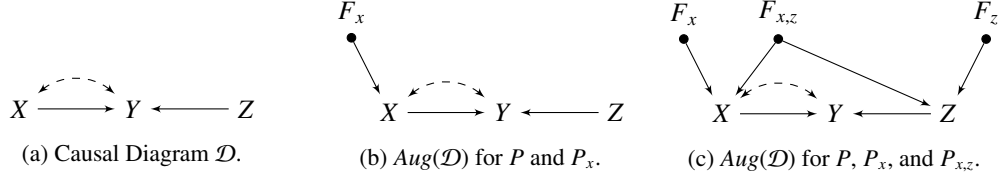

(a) Causal Diagram $\mathcal{D}$.  (b) $Aug(\mathcal{D})$ for $P$ and $P_x$.  (c) $Aug(\mathcal{D})$ for $P$, $P_x$, and $P_{x,z}$.

Figure 1: (a) Causal graph where the bidirected edge represents a latent confounder. (b) Given $P_x$, $P$, we can use $F_x$ to capture information such as *"there is a backdoor path from X to Y"* in terms of m-separation $F_x \not\perp\!\!\!\perp Y | X$. (c) Given $P$, $P_x$, $P_{x,z}$, under controlled experiment assumption, we can add $F_z$ although $P_z$ is not available. This allows us to discover that $Z$ is a cause of $Y$ and there is no confounder between them. Without adding $F_z$ this relation cannot be identified.

how a complex system works and lead to better understanding about the phenomenon under investigation. The latter, inference, attempts to leverage the causal structure to compute quantitative claims about the effect of interventions and retrospective counterfactuals, which are critical to assign credit, understand blame and responsibility, and perform judgement about fairness in decision-making.

One of the most popular languages used to encode the invariances needed to reason about causal relations, for both learning and inference, is based on graphical models, and appears under the rubric of *causal graphs* [16, 21, 2]. A causal graph is a directed acyclic graph (DAG) with latent variables, where each edge encodes a causal relationship between its endpoints: $X$ is a direct cause of $Y$, i.e., $X \rightarrow Y$, if, when the remaining factors are held constant, forcing $X$ to take a specific value affects the realization of $Y$, where $X, Y$ are random variables representing some relevant features of the system.

The task of learning the causal structure entails a search over the space of causal graphs that are compatible with the observed data; the collection of these graphs forms what is called an *equivalence class*. The most popular mark imprinted on the data by the underlying causal structure that is used to delineate an equivalence class are *conditional independence* (CI) relations. These relations are the most basic type of probabilistic invariances used in the field and have been studied at large in the context of graphical models since, at least, [15] (see also [5]). While CIs are powerful and have been the driving force behind some of the most prominent structural learning algorithms in the field [16, 21], including the PC, FCI, these are constraints specific for one distribution.

In this paper, we start by noting something very simple, albeit powerful, that happens when a combination of observational and experimental distributions are available: There are constraints over the graphical structure that emerge by comparing these different distributions, and which are not of CI-type[2]. Remarkably, and unknown until our work, the converse of the causal calculus developed by Pearl [18] offers a systematic way of reading these constraints and tying them back to the underlying graphical structure. In reference to their connection to the do-calculus rules (or a generalization, as discussed later), we call these constraints the *do-constraints*. For concreteness, consider the graph in Fig. 1(a), where the dashed-bidirected arrow represents hidden variables that generate variations of the two observed variables, $X, Y$ in this case. Suppose the observational (conditional) distribution and an interventional distribution on $X$ are available, which are written as $P(y|x)$, $P(y|do(x))$, respectively. Suppose we contrast these two distributions and the test evaluating the expression $P(y|do(x)) = P(y|x)$ comes out as *false*. This is called a do-see test since the experimental (or "do") and observational ("see") distributions are contrasted. Based on the second rule of do-calculus, one can infer that there is an open *backdoor path* from $X$ to $Y$, where the edge adjacent to $X$ on this path has an arrowhead into $X$. In our setting, we do not have access to the true graph, but we leverage this and the other do-constraints to reverse engineer the process and try to learn the structure. Broadly speaking, do-constraints will play a critical role for learning, in the same way CI/d-separation plays in learning when only observational data is available. To the best of our knowledge, this type of constraints appeared first at the very definition of causal Bayesian networks (CBNs) in [1] and then were leveraged to design efficient experiments to learn the causal graph in [12].

We assume throughout this work that interventions are *soft*. A soft intervention affects the mechanism that generates the variable, while keeping the causal connections intact. Soft-interventions are widely employed in biology and medicine, where it is hard to change the underlying system, but possibly

easier to perturb it. For our characterization, we utilize an extension of the causal calculus to soft interventions introduced in [4]. Under soft-interventions, the do-see test can be written as checking if $P_x(y|x) = P(y|x)$, where $P_x$ is the distribution obtained after a soft intervention on $X$.

The second observation leveraged here follows from another realization by Pearl that interventions can be represented explicitly in the graphical model [17]. He then introduced what we call *F-nodes*, which graphically encode the changes due to an intervention and the corresponding parametrization (see also [16, Sec. 3.2.2]). This is important in our context since the do-calculus tests will be visible more explicitly in the graph. The graph obtained by adding F-nodes to the causal graph is called the *augmented graph*. The same construct was used more prominently in [6] in the context of inference and identification. Going back to Fig. 1b, the existence of the backdoor path from $X$ to $Y$, as detected by rule 2 of the calculus, can be captured by the statement $F_X$ *is not d-separated from Y given X*. In the context of structure learning, similar constructions have been leveraged in the literature [13, 24].

We further make a specific assumption throughout the paper about the soft-interventions. We call it the *controlled experiment setting*, where each variable is intervened with the *same mechanism* change across different interventions. For example, in Fig. 1c, suppose we are given distributions from two controlled experiments $P_x, P_{x,z}$ along with observational data. We can then use $F_z$ to capture the invariances between $P_{x,z}$ and $P_x$. For example, if $P_{x,z}(y) \neq P_x(y)$, for some $y$, we can read that $F_Z \not\perp Y \mid F_X, F_{X,Z}$. Accordingly, given a set of interventional distributions, we construct an augmented graph by introducing an F-node for every unique set difference between pairs of controlled intervention sets (more on that later on). Without the controlled experiment assumption, our machinery can still be used if one knows which mechanism changes are identical and by constructing F-nodes to reflect and capture the mechanism difference across two interventions. For simplicity of presentation, however, we restrict ourselves to the controlled experiment setting and do not pursue this route explicitly.

To encapsulate the distributional invariants directly induced by the causal calculus rules[3], we call a set of interventional distributions $\mathcal{I}$-Markov to a graph, if these distributions respect the causal calculus rules relative to that graph. Note that the notion of $\mathcal{I}$-Markov is first introduced in [9, 10] for causally sufficient systems without the use of do-constraints[4]. For our characterization, we first extend the causal calculus rules to operate between arbitrary sets of interventions. We call two causal graphs $\mathcal{D}_1, \mathcal{D}_2$ $\mathcal{I}$-Markov equivalent if the set of distributions that are $\mathcal{I}$-Markov to $\mathcal{D}_1$ and $\mathcal{D}_2$ are the same. Using the augmented graph, we identify a graphical condition that is necessary and sufficient for two CBNs with latents to be $\mathcal{I}$-Markov equivalent. Finally, we propose a sound algorithm for learning the augmented graph from interventional data. Our contributions can be summarized as follows:

- We propose a characterization of $\mathcal{I}$-Markov equivalence between two causal graphs with latent variables for a given intervention set $\mathcal{I}$ that is based on a generalization of do-calculus rules to arbitrary subsets of interventions.

- We show a graphical characterization of $\mathcal{I}$-Markov equivalence of causal graphs with latents.

- We introduce a learning algorithm for inferring the graphical structure using a combination of observational and interventional data and utilizing the corresponding new constraints. This procedure comes with a new set of orientation rules. We formally show its soundness.

## 2  Background and Related Work

In this section, we introduce necessary concepts that we use throughout the paper. Upper case letters denote variables and lower case letters denote an assignment. Also, bold letters denote sets.

**Causal Bayesian Network (CBN):** Let $P(\mathbf{v})$ be a probability distribution over a set of variables $\mathbf{V}$, and let $P_{\mathbf{x}}(\mathbf{v})$ denote the distribution resulting from the *hard intervention do*$(\mathbf{X} = \mathbf{x})$, which sets $\mathbf{X} \subseteq \mathbf{V}$ to constants $\mathbf{x}$. Let $\mathbf{P}^*$ denote the set of all interventional distributions $P_{\mathbf{x}}(\mathbf{v})$, for all $\mathbf{X} \subseteq \mathbf{V}$, including $P(\mathbf{V})$. A directed acyclic graph (DAG) over $\mathbf{V}$ is said to be a *causal Bayesian network* compatible with $\mathbf{P}^*$ if and only if, for all $\mathbf{X} \subseteq \mathbf{V}$, $P_{\mathbf{x}}(\mathbf{v}) = \prod_{\{i|V_i \notin \mathbf{X}\}} P(v_i|\mathbf{pa}_i)$, for all $\mathbf{v}$ consistent with $\mathbf{x}$, and where $\mathbf{Pa}_i$ is the set of parents of $V_i$ [16, 1, pp. 24]. If so, we refer to the DAG as *causal*.

Given that a subset of the variables are unmeasured or latent, $\mathcal{D}(\mathbf{V} \cup \mathbf{L}, \mathbf{E})$ represents the causal graph where $\mathbf{V}$ and $\mathbf{L}$ denote the measured and latent variables, respectively, and $\mathbf{E}$ denotes the edges. A dashed bi-directed edge is used instead of $\leftarrow L \rightarrow$, where $L \in \mathbf{L}$, whenever $L$ is a root node with exactly two children. The observed distribution $P(\mathbf{v})$ is obtained by marginalizing $\mathbf{L}$ out.

$$P(\mathbf{v}) = \sum_{\mathbf{L}} \prod_{\{i | T_i \in \mathbf{V} \cup \mathbf{L}\}} P(t_i | \mathbf{pa}_i)$$

Clearly, the joint distribution over $\mathbf{V}$ does not factorize relative to $\mathcal{D}$ in a typical fashion, since Markovianity is no longer valid, but it does relative to both $\mathbf{V}$ and $\mathbf{L}$. Still, CI relations can be read from the graph using a graphical criterion known as *d-separation*. Also, two causal graphs are called *Markov equivalent* whenever they share the same set of conditional independences over $\mathbf{V}$.

**Soft Interventions:** Another common type of intervention is *soft*, where the original conditional distributions of the intervened variables $\mathbf{X}$ are replaced with new ones, without completely eliminating the causal effect of the parents. Accordingly, the interventional distribution $P_{\mathbf{x}}(\mathbf{v})$ becomes as follows, where $P'(X_i | Pa_i) \neq P(X_i | Pa_i)$ is the new conditional distribution set by the intervention:

$$P_{\mathbf{x}}(\mathbf{v}) = \sum_{\mathbf{L}} \prod_{\{i | X_i \in \mathbf{X}\}} P'(x_i | \mathbf{pa}_i) \prod_{\{j | T_j \notin \mathbf{X}\}} P(t_j | \mathbf{pa}_j)$$

In this work, we assume that all the soft interventions are *controlled*. This means that for any two interventions $\mathbf{I}, \mathbf{J} \subseteq \mathbf{V}$ where $X_i \in \mathbf{I} \cap \mathbf{J}$, we have $P_{\mathbf{I}}(X_i | Pa_i) = P_{\mathbf{J}}(X_i | Pa_i)$.

**Ancestral graphs:** We now introduce a graphical representation of equivalence classes of causal graphs with latent nodes. A *mixed* graph can contain directed and bi-directed edges. $A$ is an ancestor of $B$ if there is a directed path from $A$ to $B$. $A$ is a *spouse* of $B$ if $A \leftrightarrow B$ is present. If $A$ is both a spouse and an ancestor of $B$, this creates an *almost directed cycle*. An *inducing path* relative to $\mathbf{L}$ is a path on which every non-endpoint node $X \notin \mathbf{L}$ is a collider on the path (i.e., both edges incident to the node are into it) and every collider is an ancestor of an endpoint of the path. A mixed graph is *ancestral* if it does not contain a directed or almost directed cycle. It is *maximal* if there is no inducing path (relative to the empty set) between any two non-adjacent nodes. A *Maximal Ancestral Graph* (MAG) is a graph that is both ancestral and maximal [19]. Given a causal graph $\mathcal{D}(\mathbf{V}, \mathbf{L})$, a MAG $\mathcal{M}_{\mathcal{D}}$ over $\mathbf{V}$ can be constructed such that both the independence and the ancestral relations among variables in $\mathbf{V}$ are retained, see, for example, [27, p. 6].

A triple $\langle X, Y, Z \rangle$ is an unshielded triple if $X$ and $Y$ are adjacent, $Y$ and $Z$ are adjacent, and $X$ and $Z$ are not adjacent. If both edges are into $Y$, then the triple is referred to as *unshielded collider*. A path between $X$ and $Y$, $p = \langle X, \ldots, W, Z, Y \rangle$, is a *discriminating path* for $Z$ if (1) $p$ includes at least three edges; (2) $Z$ is a non-endpoint node on $p$, and is adjacent to $Y$ on $p$; and (3) $X$ is not adjacent to $Y$, and every node between $X$ and $Z$ is a collider on $p$ and is a parent of $Y$. Two MAGs are Markov equivalent if and only if (1) they have the same adjacencies; (2) they have the same unshielded colliders; and (3) if a path $p$ is a discriminating path for a vertex $Z$ in both graphs, then $Z$ is a collider on the path in one graph if and only if it is a collider on the path in the other. A *PAG*, which represents a Markov equivalence class of a MAG, is learnable from the independence model over the observed variables, and the FCI algorithm is a standard sound and complete method to learn such an object [28].

**Related Work:** Learning causal graphs from a combination of observational and interventional data has been studied in the literature [3, 11, 7, 20, 8, 12, 23]. For causally sufficient systems, the notion and characterization of interventional Markov equivalence has been introduced in [9, 10]. More recently, [24] showed that the same characterization can be used for both hard and soft interventions. For causally insufficient systems, [22] uses SAT solvers to learn a summary graph over the observed variables given data from different experimental conditions. [13] introduces an algorithm to pool experimental datasets together and runs a modification of FCI to learn an augmented graph; however, they do not consider characterizing an equivalence class.

**Notations:** For random variables $X, Y, Z$, the CI relation *$X$ is independent of $Y$ conditioned on $Z$* is shown by $X \perp\!\!\!\perp Y | Z$. The d-separation statement *node $X$ is d-separated from $Y$ given $Z$ in graph $\mathcal{D}$* is shown by $(X \perp\!\!\!\perp Y | Z)_{\mathcal{D}}$. $\mathcal{I} \subseteq 2^{\mathbf{V}}$ is reserved for a set of interventions, where $2^{\mathbf{V}}$ is the power set of $\mathbf{V}$. We show the symmetric difference by $\mathbf{I} \triangle \mathbf{J} := (\mathbf{I} \setminus \mathbf{J}) \cup (\mathbf{J} \setminus \mathbf{I})$. $\mathcal{D}_{\overline{\mathbf{X}}}$ denotes the graph obtained from $\mathcal{D}$ where all the incoming edges to the set of nodes in $\mathbf{X}$ are removed. Similarly, $\mathcal{D}_{\underline{\mathbf{X}}}$ denotes the removal of outgoing edges. We assume that there is no selection bias. A star on an endpoint of an edge $*\!\!-\!\!*$ is used as a wildcard to denote circle, arrowhead, or tail.

## 3 Do-Constraints – Combining Observational and Experimental Distributions

One of the most celebrated results in causal inference comes under the rubric of do-calculus (or causal calculus) [18, 16]. The calculus consists of a set of inference rules that allows one to create a map between distributions generated by a causal graph when certain graphical conditions hold in the graph. The calculus was developed in the context of hard interventions, and recent work presented a generalization of this result for soft interventions [4], which we state next:

**Theorem 1** (Special case of Thm. 1 in [4]). *Let $\mathcal{D} = (\mathbf{V} \cup \mathbf{L}, \mathbf{E})$ be a causal graph. Then, the following holds for any strictly positive distribution consistent with $\mathcal{D}$.*

*Rule 1 (see-see): For any $\mathbf{X} \subseteq \mathbf{V}$ and disjoint $\mathbf{Y}, \mathbf{Z}, \mathbf{W} \subseteq \mathbf{V}$*

$$P_x(y|w, z) = P_x(y|w) \qquad \text{if } Y \perp\!\!\!\perp Z \,|W \text{ in } \mathcal{D}.$$

*Rule 2 (do-see): For any disjoint $\mathbf{X}, \mathbf{Y}, \mathbf{Z} \subseteq \mathbf{V}$ and $\mathbf{W} \subset \mathbf{V} \setminus (\mathbf{Z} \cup \mathbf{Y})$*

$$P_{x,z}(y|z, w) = P_x(y|z, w) \qquad \text{if } Y \perp\!\!\!\perp Z \,|W \text{ in } \mathcal{D}_{\underline{Z}}.$$

*Rule 3 (do-do): For any disjoint $\mathbf{X}, \mathbf{Y}, \mathbf{Z} \subseteq \mathbf{V}$ and $\mathbf{W} \subset \mathbf{V} \setminus (\mathbf{Z} \cup \mathbf{Y})$*

$$P_{x,z}(y|w) = P_x(y|w) \qquad \text{if } Y \perp\!\!\!\perp Z \,|W \text{ in } \mathcal{D}_{\overline{Z(W)}},$$

*where $\mathbf{Z(W)} \subseteq \mathbf{Z}$ are non-ancestors of $\mathbf{W}$ in $\mathcal{D}$.*

The first rule of the calculus is a d-separation type of statement relative to a specific interventional distribution $P_x$, which says that $Y \perp\!\!\!\perp Z \,|W$ in $\mathcal{D}$ implies the corresponding conditional independence $P_x(y|w, z) = P_x(y|w)$. Note that the converse of this rule is the work horse underlying most of the structure learning algorithms found in practice, which says that if some independence hold in $P$, this would imply a corresponding graphical separation (under faithfulness). In the case just mentioned, this would imply that $Y$ and $Z$ should be separated in $\mathcal{D}$, meaning, they have neither a directed nor a bidirected arrow connecting them.

From this understanding, we make a very simple, albeit powerful observation – i.e., the converse of the other two rules should offer insights about the underlying graphical structure as well. To witness, consider the causal graph $\mathcal{D} = \{X \rightarrow Y, X \dashleftarrow\dashrightarrow Y\}$, and suppose we have the observational and interventional distributions $P(Y, X)$ and $P_X(Y, X)$, respectively. Using the CI tests $P(Y, X) \neq P(Y) \cdot P(X)$ and $P_X(Y, X) \neq P_X(Y) \cdot P_X(X)$, we infer that the two variables are dependent (or not independent) and consequently d-connected in the graph, while no claim can be made about the causal relation between them. Given the inequality $P_X(Y) \neq P(Y)$, we infer that the condition for rule 3 does not hold and $Y \not\perp\!\!\!\perp X$ in $\mathcal{D}_{\overline{X}}$. Hence, $X$ must be a cause of $Y$ – changing the value of $X$ has a downstream effect on $Y$. Similarly, given the inequality $P_X(Y|X) \neq P(Y|X)$, the condition related to rule 2 does not hold, and $Y \not\perp\!\!\!\perp X$ in $\mathcal{D}_{\underline{X}}$. The implication in this case is that there is an unblockable backdoor path between $X$ and $Y$ that is into $X$, i.e., a latent variable. Alternatively, if $\mathcal{D} = \{X \rightarrow Y\}$, then $P_X(Y|X) = P(Y|X)$, under faithfulness, implies the absence of a latent variable by the converse of rule 2.

Broadly speaking, rule 3 allows one to infer causal relations between variables, and consequently directed edges in the causal graph. Since the compared interventional distributions differ by a subset of interventions ($Z$), we call this the *do-do* test. On the other hand, rule 2 allows one to infer spurious relations between variables, and consequently latent variables in the causal graph[5]. The *do-see* naming of the test stems from the fact that we compare a distribution with an intervention on a subset $Z$ (do) versus another which only conditions on $Z$ (see). Naturally, rule 1 is the usual conditional independence test that allows one to detect that neither directed nor bidirected arrow exists.

Putting together these rules, we show in Corollary 1 a generalization of rules 2 and 3 . Note that rule 2 appears when $\mathbf{J} \subset \mathbf{I}$ and $\mathbf{I} \setminus \mathbf{J} \subseteq \mathbf{W}$; similarly, rule 3 can be seen when $\mathbf{J} \subset \mathbf{I}$ and $(\mathbf{I} \setminus \mathbf{J}) \cap \mathbf{W} = \emptyset$.

**Corollary 1** (mixed do-do/do-see). *Let $\mathcal{D} = (\mathbf{V} \cup \mathbf{L}, \mathbf{E})$ be a causal graph. Under the controlled intervention assumption, for any $\mathbf{I}, \mathbf{J} \subseteq \mathbf{V}$ and disjoint $\mathbf{Y}, \mathbf{W} \subseteq \mathbf{V}$, we have the following:*

$$P_{\mathbf{I}}(\mathbf{y}|\mathbf{w}) = P_{\mathbf{J}}(\mathbf{y}|\mathbf{w}) \qquad \text{if } \mathbf{Y} \perp\!\!\!\perp \mathbf{K} \,|\mathbf{W} \setminus \mathbf{W_k} \text{ in } \mathcal{D}_{\underline{\mathbf{W_k}}, \overline{\mathbf{R(W)}}},$$

*where $\mathbf{K} := \mathbf{I} \triangle \mathbf{J}$, $\mathbf{W_k} =: \mathbf{W} \cap \mathbf{K}$, $\mathbf{R} := \mathbf{K} \setminus \mathbf{W_k}$, and $\mathbf{R(W)} \subseteq \mathbf{R}$ are non-ancestors of $\mathbf{W}$ in $\mathcal{D}$.*

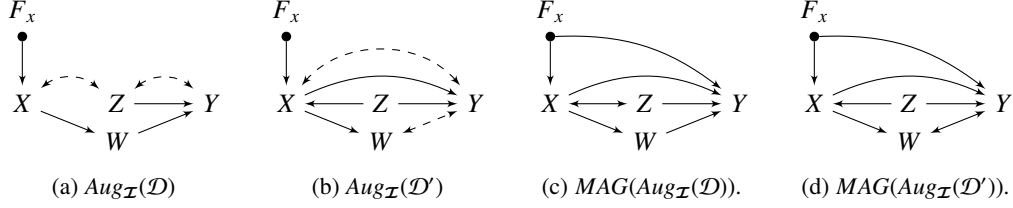

(a) $Aug_{\mathcal{I}}(\mathcal{D})$      (b) $Aug_{\mathcal{I}}(\mathcal{D}')$      (c) $MAG(Aug_{\mathcal{I}}(\mathcal{D}))$.      (d) $MAG(Aug_{\mathcal{I}}(\mathcal{D}'))$.

Figure 2: Augmented graphs with respect to $\mathcal{I} = \{\emptyset, \{X\}\}$ and the corresponding augmented MAGs.

In general, the proposed rule is a mixture of rules 2 and 3 as we could be conditioning in $\mathbf{W}$ on a subset of the symmetrical difference set $\mathbf{I} \triangle \mathbf{J}$. For instance, consider the causal graph $\mathcal{D} = \{C \longleftrightarrow A \to B, C \longleftrightarrow B\}$ and suppose we have the interventional distributions $P_{A,B}$ and $P_{C,B}$. Since $B \perp\!\!\!\perp \{A, C\}$ in $\mathcal{D}_{\underline{A},\overline{C}}$, then $P_{A,B}(B|A) = P_{B,C}(B|A)$. This generalization will soon play a significant role in the characterization and learning of the interventional equivalence class.

# 4   Interventional Markov Equivalence under Do-constraints

In this section, the new do-constraints will be used to define the notion of interventional Markov equivalence. Then, we will characterize when two causal graphs are equivalent in accordance to the proposed definition. We start by defining the notion of interventional Markov as shown below.

**Definition 1.** *Consider the tuples of absolutely continuous probability distributions $(P_{\mathbf{I}})_{\mathbf{I} \in \mathcal{I}}$ over a set of variables $\mathbf{V}$. A tuple $(P_{\mathbf{I}})_{\mathbf{I} \in \mathcal{I}}$ satisfies the $\mathcal{I}$-Markov property with respect to a graph $\mathcal{D} = (\mathbf{V} \cup \mathbf{L}, \mathbf{E})$ if the following holds for disjoint $\mathbf{Y}, \mathbf{Z}, \mathbf{W} \subseteq \mathbf{V}$:*

*(1) For $\mathbf{I} \in \mathcal{I}$:*      $P_{\mathbf{I}}(\mathbf{y}|\mathbf{w}, \mathbf{z}) = P_{\mathbf{I}}(\mathbf{y}|\mathbf{w})$      *if $\mathbf{Y} \perp\!\!\!\perp \mathbf{Z} | \mathbf{W}$ in $\mathcal{D}$.*

*(2) For $\mathbf{I}, \mathbf{J} \in \mathcal{I}$:*      $P_{\mathbf{I}}(\mathbf{y}|\mathbf{w}) = P_{\mathbf{J}}(\mathbf{y}|\mathbf{w})$      *if $\mathbf{Y} \perp\!\!\!\perp \mathbf{K} | \mathbf{W} \setminus \mathbf{W_k}$ in $\mathcal{D}_{\underline{\mathbf{W_k}}, \overline{\mathbf{R(W)}}}$,*

*where $\mathbf{K} := \mathbf{I} \triangle \mathbf{J}$, $\mathbf{W_k} =: \mathbf{W} \cap \mathbf{K}$, $\mathbf{R} := \mathbf{K} \setminus \mathbf{W_k}$, and $\mathbf{R(W)} \subseteq \mathbf{R}$ are non-ancestors of $\mathbf{W}$ in $\mathcal{D}$.*

*The set of all tuples that satisfy the $\mathcal{I}$-Markov property with respect to $\mathcal{D}$ are denoted by $\mathcal{P}_{\mathcal{I}}(\mathcal{D}, \mathbf{V})$.*

The two conditions used in the definition correspond to rule 1 of Theorem 1 and that of Corollary 1. Notice that the traditional Markov definition only considers the first condition over the observational distribution $P(\mathbf{V})$; a case included in the $\mathcal{I}$-Markov whenever $\emptyset \in \mathcal{I}$. Accordingly, two causal graphs are said to be $\mathcal{I}$-Markov equivalent if they license the same set of distribution tuples. This notion is formalized in the following definition.

**Definition 2.** *Given two causal graphs $\mathcal{D}_1 = (\mathbf{V} \cup \mathbf{L}_1, \mathbf{E}_1)$ and $\mathcal{D}_2 = (\mathbf{V} \cup \mathbf{L}_2, \mathbf{E}_2)$, and an intervention set $\mathcal{I} \subseteq 2^{\mathbf{V}}$, $\mathcal{D}_1$ and $\mathcal{D}_2$ are called $\mathcal{I}$-Markov equivalent if $\mathcal{P}_{\mathcal{I}}(\mathcal{D}_1, \mathbf{V}) = \mathcal{P}_{\mathcal{I}}(\mathcal{D}_2, \mathbf{V})$.*

One challenge with Definition 1 is that testing for the d-separation statement in condition (2) requires a mutilated graph where we cut some of the edges in $\mathcal{D}$. This makes it harder to represent all the constraints imposed by a causal graph compactly. Accordingly, we use the notion of an *augmented graph* that is introduced below (Definition 3). In words, the construction of the augmented graph goes as follows. First, initialize the augmented graph to the input causal graph. Then, for every distinct symmetric set difference between $\mathbf{I}, \mathbf{J} \in \mathcal{I}$, denoted by $\mathbf{S}_i$, introduce a new node $F_i$ and make it a parent to each node in $\mathbf{S}_i$, i.e., $F_i \to S \in \mathbf{S}_i$. Note that this type of construction has been used in the literature to model interventions [17, 6]. For example, for $\mathcal{I} = \{\emptyset, \{X\}\}$, Figure 2a presents the augmented graph corresponding to the causal graph, which is the induced subgraph over $\{X, W, Z, Y\}$. Node $F_x$ is added in accordance with the symmetrical difference set $(\emptyset \setminus \{X\}) \cup (\{X\} \setminus \emptyset) = \{X\}$.

**Definition 3** (Augmented graph). *Consider a causal graph $\mathcal{D} = (\mathbf{V} \cup \mathbf{L}, \mathbf{E})$ and an intervention set $\mathcal{I} \subseteq 2^{\mathbf{V}}$. Let $\mathcal{S} = \{\mathbf{S}_1, \mathbf{S}_2, \dots, \mathbf{S}_k\} = \{S : \exists \mathbf{I}, \mathbf{J} \in \mathcal{I} \text{ s.t. } \mathbf{I} \triangle \mathbf{J} = S\}$. The augmented graph of $\mathcal{D}$ with respect to $\mathcal{I}$, denoted as $Aug_{\mathcal{I}}(\mathcal{D})$, is the graph constructed as follows: $Aug_{\mathcal{I}}(\mathcal{D}) = (\mathbf{V} \cup \mathcal{F}, \mathbf{E} \cup \mathcal{E})$ where $\mathcal{F} := \{F_i\}_{i \in [k]}$ and $\mathcal{E} = \{(F_i, j)\}_{i \in [k], j \in \mathbf{S}_i}$.*

The significance of the augmented graph construction is illustrated by Proposition 1, which provides criteria to test the d-separation statements in Definition 1 equivalently from the corresponding augmented graph of a causal graph. Back to the example in Figure 2a, the statement $Y \perp\!\!\!\perp X | Z$ in

$\mathcal{D}_{\overline{X}}$ can be equivalently tested by the statement $Y \perp\!\!\!\perp F_x | Z$ in the corresponding augmented graph. Similarly, $Y \perp\!\!\!\perp X$ in $\mathcal{D}_{\underline{X}}$ can be equivalently tested by $Y \perp\!\!\!\perp F_x | X$ in $Aug_{\mathcal{I}}(\mathcal{D})$.

**Proposition 1.** *Consider a causal graph $\mathcal{D} = (\mathbf{V} \cup \mathbf{L}, \mathbf{E})$ and the corresponding augmented graph $Aug_{\mathcal{I}}(\mathcal{D}) = (\mathbf{V} \cup \mathbf{L} \cup \mathcal{F}, \mathbf{E} \cup \mathcal{E})$ with respect to an intervention set $\mathcal{I}$, where $\mathcal{F} = \{F_i\}_{i \in [k]}$. Let $\mathbf{S}_i$ be the set of nodes adjacent to $F_i, \forall i \in [k]$. We have the following equivalence relations.*

*For disjoint $\mathbf{Y}, \mathbf{Z}, \mathbf{W} \subseteq \mathbf{V}$:*

$$(\mathbf{Y} \perp\!\!\!\perp \mathbf{Z} | \mathbf{W})_{\mathcal{D}} \iff (\mathbf{Y} \perp\!\!\!\perp \mathbf{Z} | \mathbf{W}, F_{[k]})_{Aug(\mathcal{D})} \tag{1}$$

*For disjoint $\mathbf{Y}, \mathbf{W} \subseteq \mathbf{V}$, where $\mathbf{W}_i := \mathbf{W} \cap \mathbf{S}_i, \mathbf{R} := \mathbf{S}_i \setminus \mathbf{W}_i$:*

$$(\mathbf{Y} \perp\!\!\!\perp \mathbf{S}_i | \mathbf{W} \setminus \mathbf{W}_i)_{\mathcal{D}_{\mathbf{W}_i, \overline{\mathbf{R}(\mathbf{W})}}} \iff (\mathbf{Y} \perp\!\!\!\perp F_i | \mathbf{W}, F_{[k] \setminus \{i\}})_{Aug(\mathcal{D})} \tag{2}$$

In order to characterize causal graphs that are $\mathcal{I}$-Markov equivalent, we draw some insight from the Markov equivalence of causal graphs with latents. Ancestral graphs, and more specifically MAGs, were proposed as a representation to encode the d-separation statements of a causal graph among the measured variables while not explicitly encoding the latent nodes. The definition below (Def. 4) introduces the *augmented MAG* that is constructed over an augmented graph. Since all the constraints in the $\mathcal{I}$-Markov definition can be tested by d-separation statements in the augmented graph, then an augmented MAG preserves all those constraints. For example, Figs. 2c and 2d present the augmented MAGs corresponding to the augmented graphs in Figs. 2a and 2b, respectively. Notice that $F_x$ and $Y$ are adjacent in both MAGs since they are not separable by any set in the augmented graphs.

**Definition 4** (Augmented MAG). *Given a causal graph $\mathcal{D} = (\mathbf{V} \cup \mathbf{L}, \mathbf{E})$ and an intervention set $\mathcal{I}$, the augmented MAG is the MAG constructed over $\mathbf{V}$ from $Aug_{\mathcal{I}}(\mathcal{D})$, i.e., $MAG(Aug_{\mathcal{I}}(\mathcal{D}))$.*

Below, we derive a characterization for two causal graphs to be $\mathcal{I}$-Markov equivalent – two causal graphs are $\mathcal{I}$-Markov equivalent if their corresponding augmented MAGs satisfy the three conditions given in Theorem 2. For example, the two augmented MAGs in Figures 2c and 2d satisfy the three conditions, hence the original causal graphs are in the same $\mathcal{I}$-Markov equivalence class.

**Theorem 2.** *Two causal graphs $\mathcal{D}_1 = (\mathbf{V} \cup \mathbf{L}_1, \mathbf{E}_1)$ and $\mathcal{D}_2 = (\mathbf{V} \cup \mathbf{L}_2, \mathbf{E}_2)$ are $\mathcal{I}$-Markov equivalent for a set of controlled experiments $\mathcal{I}$ if and only if for $\mathcal{M}_1 = MAG(Aug_{\mathcal{I}}(\mathcal{D}_1))$ and $\mathcal{M}_2 = MAG(Aug_{\mathcal{I}}(\mathcal{D}_2))$:*

1. *$\mathcal{M}_1$ and $\mathcal{M}_2$ have the same skeleton;*

2. *$\mathcal{M}_1$ and $\mathcal{M}_2$ have the same unshielded colliders;*

3. *If a path $p$ is a discriminating path for a node $Y$ in both $\mathcal{M}_1$ and $\mathcal{M}_2$, then $Y$ is a collider on the path in one graph if and only if it is a collider on the path in the other.*

## 5   Learning by Combining Observations and Experiments

In this section, we develop an algorithm to learn the augmented graph from a combination of observational and interventional data, which consequently recovers the causal graph. However, similar to the observational case, it is typically impossible to completely determine the causal graph from the available measured data, especially when latents are present. Then, the objective is to learn a class of augmented MAGs consistent with data. For this, we define an augmented PAG as follows.

**Definition 5.** *Given a causal graph $\mathcal{D}$ and an intervention set $\mathcal{I}$, let $\mathcal{M} = MAG(Aug_{\mathcal{I}}(\mathcal{D}))$ and let $[\mathcal{M}]$ be the set of augmented MAGs corresponding to all the causal graphs that are $\mathcal{I}$-Markov equivalent to $\mathcal{D}$. An Augmented PAG for $\mathcal{D}$, denoted $\mathcal{G} = PAG(Aug_{\mathcal{I}}(\mathcal{D}))$, is a graph such that:*

1. *$\mathcal{G}$ has the same adjacencies as $\mathcal{M}$, and any member of $[\mathcal{M}]$ does; and*

2. *every non-circle mark in $\mathcal{G}$ is an invariant mark in $[\mathcal{M}]$.*

As with any learning algorithm, some faithfulness assumption is needed to infer graphical properties from the corresponding distributional constraints. Hence, we assume that the given interventional distributions are *c-faithful* to the causal graph $\mathcal{D}$ as defined below.

---

**Algorithm 1** Algorithm for Learning Augmented PAG

---

1: **function** LearnAugPAG($\mathcal{I}, (P_\mathbf{I})_{\mathbf{I}\in\mathcal{I}}, \mathbf{V}$)
2:    $(\mathcal{F}, \mathcal{S}, \sigma) \leftarrow$ CreateAugmentedNodes($\mathcal{I}, \mathbf{V}$)
3:    $\mathbf{V} \leftarrow \mathbf{V} \cup \mathcal{F}$
4:    **Phase I: Learn Adjacencies and Seperating Sets**
5:    Form the complete graph $\mathcal{G}$ on $\mathbf{V}$ where between every pair of nodes there is an edge ∘–∘.
6:    **for** Every pair $X, Y \in \mathbf{V}$ **do**
7:        **if** $X \in \mathcal{F} \wedge Y \in \mathcal{F}$ **then**
8:            $SepSet(X, Y) \leftarrow \emptyset, SepFlag(X, Y) = True$
9:        **else**
10:            $(SepSet(X, Y), SepFlag) \leftarrow$ Do-Constraints($(P_\mathbf{I})_{\mathbf{I}\in\mathcal{I}}, X, Y, V, \mathcal{F}, \sigma$)
11:        **if** $SepFlag = True$ **then**
12:            Remove the edge between $X, Y$ in $\mathcal{G}$.
13:    **Phase II: Learn Unshielded Colliders**
14:    For every unshielded triple $\langle X, Z, Y \rangle$ in $\mathcal{G}$, orient it as $X*\!\!\rightarrow Z \leftarrow\!\!*Y$ iff $Z \notin SepSet(X, Y)$
15:    **Phase III: Apply Orientation Rules**
16:    Apply 7 FCI rules in [28] together with the following 2 additional rules until none applies.
17:    *Rule 8:* For any $F_k \in \mathcal{F}$, orient adjacent edges out of $F_k$.
18:    *Rule 9:* For any $F_k \in \mathcal{F}$ that is adjacent to a node $Y \notin \mathbf{S}_k$
19:        If $|\mathbf{S}_k| = 1$, orient $X *\!\!-\!\!* Y$ as $X \rightarrow Y$ for $X \in \mathbf{S}_k$.

---

---

**Algorithm 2** Creating F-nodes.

---

1: **function** CreateAugmentedNodes($\mathcal{I}, \mathbf{V}$)
2:    $\mathcal{F} = \emptyset, \mathcal{S} = \emptyset, k = 0, \sigma : \mathbb{N} \rightarrow 2^\mathbf{V} \times 2^\mathbf{V}$
3:    **for** all pairs $\mathbf{I}, \mathbf{J} \in \mathcal{I}$, if $\mathbf{I} \triangle \mathbf{J} \notin \mathcal{S}$ **do**
4:        Set $k \leftarrow k + 1$, set $\mathbf{S}_k = \mathbf{I} \triangle \mathbf{J}$, add $F_k$ to $\mathcal{F}$, add $\mathbf{S}_k$ to $\mathcal{S}$, set $\sigma(k) = (\mathbf{I}, \mathbf{J})$.
    **return** $\mathcal{F}, \mathcal{S}, \sigma$

---

**Definition 6.** *Consider a causal graph $\mathcal{D} = (\mathbf{V} \cup \mathbf{L}, \mathbf{E})$. A tuple of distributions $(P_\mathbf{I})_{\mathbf{I}\in\mathcal{I}} \in \mathcal{P}(\mathcal{D}, \mathbf{V})$ is called c-faithful to graph $\mathcal{D}$ if the converse for each of the conditions given in Definition 1 holds.*

Algorithm 1 presents a modification of the FCI algorithm to learn augmented PAGs. To explain the algorithm, we first describe FCI which, given an independence model over the measured variables, proceeds in three phases [25]: In phase I, the algorithm initializes a complete graph with circle edges (∘–∘), then it removes the edge between any pair of nodes if a separating set between the pair exists and records the set. In phase II, the algorithm identifies unshielded triples $\langle A, B, C \rangle$ and orients the edges into $B$ if $B$ is not in the separating set of $A$ and $C$. Finally, in phase III, FCI applies the orientation rules. Only one of the rules uses separating sets while the rest use MAG properties, and soundness and completeness of the previous phases – the skeleton is correct and all the unshielded colliders are discovered. We note that FCI looks for any separating sets, and not necessarily the minimal ones. We also observe that if two nodes $X, Y$ are separated given $\mathbf{Z}$ in $Aug_\mathcal{I}(\mathcal{D})$, they are also separated given $\mathbf{Z} \cup \mathcal{F}$ since $\mathcal{F}$ are root nodes by construction, i.e., all the edges incident on F-nodes are out of them.

Algorithm 1 follows a similar flow to that of the FCI. In phase I, it learns the skeleton of the augmented PAG. Function CreateAugmentedNodes($\cdot$) in Alg. 2 creates the F-nodes by computing the set $\mathcal{S}$ of unique symmetric difference sets from all pairs of interventions in $\mathcal{I}$. Sigma ($\sigma$) maps every F-node to a source pair of interventions, which is used later on to perform the do-tests. The algorithm starts by creating a complete graph of circle edges between $\mathbf{V} \cup \mathcal{F}$. Then, it removes the edge between any two nodes $X$ and $Y$ if a separating set exists. If the two nodes are F-nodes, then they are separated by the empty set by construction. Otherwise, it calls the function Do-Constraints($\cdot$) in Alg. 3 to search for a separating set using the corresponding do-constraints. The function routine works as follows: If the two nodes are random variables (and not F-nodes), then an arbitrary distribution is chosen and we find a subset $\mathbf{W}$ that establishes conditional independence between $X$ and $Y$ (rule 1 of Thm. 1). Else, one of the two nodes is an F-node; without loss of generality, we choose it to be $X$. The algorithm then looks for a subset $\mathbf{W}$ that satisfies the invariance of Corollary 1, i.e., $P_\mathbf{I}(y|\mathbf{w}) = P_\mathbf{J}(y|\mathbf{w})$.

Phase II of Alg. 1 is similar to the FCI counterpart. For the edge orientation phase, note that the augmented MAG is a MAG indeed, hence all the FCI orientation rules still apply. Therefore, phase III

---

**Algorithm 3** Find m-separation sets via Calculus Tests.

---

 1: **function** Do-Constraints($\mathcal{I}, (P_{\mathbf{I}})_{\mathbf{I}\in\mathcal{I}}, X, Y, \mathbf{V}, \mathcal{F}, \sigma$)
 2:     $SepSet = \emptyset, SepFlag = False$
 3:     **if** $X \notin \mathcal{F} \wedge Y \notin \mathcal{F}$ **then**
 4:         Pick $\mathbf{I} \in \mathcal{I}$ arbitrarily.
 5:         **for** $\mathbf{W} \subseteq \mathbf{V} \setminus \mathcal{F}$ **do**
 6:             **if** $P_{\mathbf{I}}(y|\mathbf{w}, x) = P_{\mathbf{I}}(y|\mathbf{w})$ **then** $SepSet = \mathbf{W} \cup \mathcal{F}, SepFlag = True$
 7:     **else**
 8:         Suppose $X \in \mathcal{F}, Y \notin \mathcal{F}$ and $X = F_i$ without loss of generality.
 9:         $(\mathbf{I}, \mathbf{J}) = \sigma(i)$
10:         **for** $\mathbf{W} \subseteq \mathbf{V} \setminus (\mathcal{F} \cup \mathbf{Y})$ **do**
11:             **if** $P_{\mathbf{I}}(y|\mathbf{w}) = P_{\mathbf{J}}(y|\mathbf{w})$ **then** $SepSet = \mathbf{W}, \mathcal{F} \setminus \{F_i\}, SepFlag = True$
        **return** $(SepSet, SepFlag)$

---

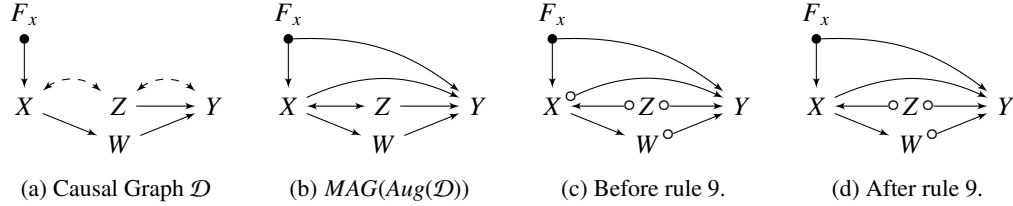

(a) Causal Graph $\mathcal{D}$      (b) $MAG(Aug(\mathcal{D}))$      (c) Before rule 9.      (d) After rule 9.

Figure 3: An example of learning the augmented PAG from the distributions $P, P_x$ consistent with the given causal graph. Rule 9 allows orienting the tail at $X \circ\!\!\rightarrow Y$.

uses the FCI orientation rules along with the following two new ones. The algorithm keeps applying the rules until none applies anymore.

**Rule 8 (F-node Edges):** For any edge adjacent to an $F$ node, orient the edge out of the $F$ node.

**Rule 9 (Inducing Paths):** If $F_k \in \mathcal{F}$ is adjacent to a node $Y \notin \mathbf{S}_k$ and $|\mathbf{S}_k| = 1$, e.g., $\mathbf{S}_k = \{X\}$, then orient $X \ast\!\!-\!\!\ast Y$ out of $X$, i.e., $X \rightarrow Y$. The intuition for this rule is as follows: If $F_k$ is adjacent to a node $Y \notin \mathbf{S}_k$ in $\mathcal{G}$, then there is an inducing path $p$ between $F_k$ and $Y$ in $Aug_{\mathcal{I}}(\mathcal{D})$, where $\mathcal{D}$ is any causal graph in the equivalence class. Since $F_k$ is a root node and by the properties of inducing paths, the subpath of $p$ from $X$ to $Y$ is an inducing path as well and $X$ is an ancestor of $Y$ in $Aug_{\mathcal{I}}(\mathcal{D})$. Hence, the edge between $X$ and $Y$ is out of $X$ and into $Y$ in $MAG(Aug_{\mathcal{I}}(\mathcal{D}))$ and consequently in $\mathcal{G}$.

We give an example to illustrate the steps of the algorithm in Figure 3, where $\mathcal{I} = \{\emptyset, \{X\}\}$. Figure 3a shows the augmented causal graph, i.e., $Aug_{\mathcal{I}}(\mathcal{D})$, and Figure 3b shows the corresponding augmented MAG, i.e., $MAG(Aug_{\mathcal{I}}(\mathcal{D}))$. Nodes $F_x$ and $Z$ are separable in $Aug_{\mathcal{I}}(\mathcal{D})$ given the empty set and this can be tested by the do-constraint $P(Z) = P_X(Z)$. Similarly, we can infer the separation of $F_x$ and $W$ by the test $P(W|X) = P_X(W|X)$. Figure 3c shows the graph obtained after applying the seven rules of the FCI together with Rule 8. Finally, by applying Rule 9, we infer that the edge between $X$ and $Y$ has a tail at $X$ and we obtain the graph in Figure 3d. The soudness of the algorithm is shown next.

**Theorem 3.** *Consider a set of interventional distributions $(P_{\mathbf{I}})_{\mathbf{I}\in\mathcal{I}}$ c-faithful to a causal graph $\mathcal{D} = (\mathbf{V} \cup \mathbf{L})$, where $\mathcal{I}$ is a set of controlled experiments. Algorithm 1 is sound, i.e., every adjacency and orientation is common for all $MAG(Aug(\mathcal{D}'))$ where $\mathcal{D}'$ is $\mathcal{I}$-Markov equivalent to $\mathcal{D}$.*

## 6   Conclusions

We investigate the problem of learning the causal structure underlying a phenomenon of interest from a combination of observational and experimental data. We pursue this endeavor by noting that a generalization of the converse of Pearl's do-calculus (Thm. 1) leads to new tests that can be evaluated against data. These tests, in turn, translate into constraints over the structure itself. We then define an interventional equivalence class based on such criteria (Def. 1), and then derive a graphical characterization for the equivalence of two causal graphs (Thm. 2). Finally, we develop an algorithm to learn an interventional equivalence class from data, which includes new orientation rules.

## Acknowledgements

Bareinboim and Jaber are supported in parts by grants from NSF IIS-1704352, IIS-1750807 (CA-REER), IBM Research, and Adobe Research. Kocaoglu and Shanmugam are supported by the MIT-IBM Watson AI Lab.

## Footnotes

[2]Recall that a CI represents a constraint readable from one specific distribution saying that the value of $Z$ is irrelevant for computing the likelihood of $Y$ once we know the value of $X$, i.e., $P(Y|X,Z) = P(Y|X), \forall X, Y, Z$.

[3]There may be constraints that can be obtained by applying the rules multiple times we do not consider here.

[4]In the causally sufficient case, name is in reference to both global and local Markov conditions. However, in our work, the name stems from our observation that the do-constraints correspond to the global Markov conditions in the augmented graph.

[5] More precisely, rule 2 allows us to detect inducing paths that are into both variables.

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
