[Supplementary Material]

# 7 Appendix

## 7.1 Do-calculus rules for soft interventions

A recent work developed an extension of the do-calculus rules to soft interventions in structural causal models (SCMs) [4]. We reproduce a variation of this result for CBNs for completeness.

*Proof of Theorem 1.* Note that a soft-intervention does not change the underlying causal graph. Since interventional distribution factorizes with respect to the original graph, any m-separation statement in graph $D$ implies conditional independence. Under the strict positivity, conditional independence is equivalent to the invariance given in the rule, which concludes the proof.

For the proof of next two rules, similar to [18], we introduce F-nodes as random variables. Notice that this is different than the augmented graph construction we have in the main text, where we treat F-nodes as parameters. This is allowed as only a single F-node is introduced to show the result here, which is explained next.

Construct the probability distribution $p^*$ on $V \cup \{F\}$ as follows: $p^*(V|F = 0) = p_{x,z}(V), p^*(V|F = 1) = p_x(V)$, where $p_{x,z}$ is the interventional distribution after a soft intervention on the set $X \cup Z$ and $p_x$ is the interventional distribution after a soft intervention on the set $X$ is performed. Marginal distribution of $p^*(F)$ can be picked arbitrarily from the set of strictly positive distribuitons for our purposes. Assume that interventions are controlled, i.e., $p_{x,z}(x|pa_x) = p_x(x|pa_x)$, where $pa_x$ is the set of parents of node $X$.

The desired equality in Rule 2 can be rewritten as $p^*(y|z, w, F = 0) = p^*(y|z, w, F = 1)$. Under the assumption of strictly positive distributions, this invariance is implied by the conditional independence statements $(Y \perp\!\!\!\perp Z |W)_{p^*}$. Therefore, we need to show that the graph separation statement given in the rule implies the desired conditional independence statement.

For this, observe that $p^*$ can be factorized as follows:

$$p^*(V, F) = p^*(F)p^*(V|F) = p^*(F)p^*(z|pa_z, F) \prod_{u \neq z} p(u|pa_u).$$  (3)

where $pa_x$ are the parents of $x$ in $D$. Note that in $G$ the set of parents of $Z$ is $pa_Z \cup F$. Therefore, $p^*$ factorizes according to the graph $G$. This implies that any d-separation statement on $G$ implies conditional independence [16][Theorem 1.2.4]. Therefore, we only need to show that the separation statement given in the rule on mutilated graph implies d-separation statement between $F_z$ and $Y$ given $Z, W$.

If $Y \perp\!\!\!\perp Z |W$ in $D_{\underline{Z}}$, this means there is no backdoor path from $Z$ to $Y$ that is active conditioned on $W$. Since $F_z$ only has an edge into $Z$, conditioned on $W, Z$ any d-connecting path to $Y$ must go through a backdoor from $Z$. However the statement $Y \perp\!\!\!\perp Z |W$ in $D_{\underline{Z}}$ implies this cannot happen, implying that $F_z \perp\!\!\!\perp Y |Z, W$ in $G$, completing the proof.

For the proof of rule 3, we use a similar argument under strict positivity. Consider the same $p^*$ construction. Similarly, this distribution factorizes with respect to graph $G$ which means and d-separation statement implies conditional independence. Therefore we only need to show that the given separation statement in the mutilated graph implies the desired d-separation statement in $G$. Suppose $Y \perp\!\!\!\perp Z |W$ in $D_{\overline{Z(W)}}$. This implies that given $W$, there is no active path from the nodes in $Z - Z(W)$ to $Y$. Moreover there is no front-door path from the elements of $Z(W)$ to $Y$ given $W$. Suppose for the sake of contradiction that $F_Z \not\perp\!\!\!\perp Y |W$ in $G$. Since $F_Z$ only has edges into $Z$, any active path must go through an element in $Z$. Suppose it goes through an element in $Z(W)$. Since no descendant of $Z(W)$ is conditioned on, the active path must go through a backdoor in $Z(W)$. However this would imply $Y \not\perp\!\!\!\perp Z |W$ in $D_{\overline{Z(W)}}$, which leads to contradiction. Now suppose active path goes through an element in $Z - Z(W)$. However, these nodes are not mutilated in $G$, hence the same active path would persist in $D$ as well, contradicting with the statement $Y \perp\!\!\!\perp Z |W$ in $D_{\overline{Z(W)}}$. Therefore we have $F_Z \perp\!\!\!\perp Y |W$ in $G$ which concludes the proof. □

## 7.2 Generalized Do-calculus Rules

In this section, we extend the do-calculus rules to be able to apply them across two arbitrary interventions. This is essential for the characterizing of our equivalence class, when arbitrary sets of interventional distributions are available.

**Proposition 2** (Generalized do-calculus for soft interventions). *Let $(D = (V \cup L, E), p)$ be a CBN with latents. Then for any set of strictly positive soft-interventional distributions $\{p_I\}_{I \in \mathcal{I}}, \mathcal{I} \subseteq 2^V$ the following holds.*

Rule 1 (conditional independence): *For any $I \subseteq V$ and disjoint $Y, Z, W \subseteq V$*

$$p_I(y|w, z) = p_I(y|w) \text{ if } Y \perp\!\!\!\perp Z | W \text{ in } D. \tag{4}$$

Rule 2 (do-see): *For any $I, J \subseteq V$ and disjoint $Y, W \subseteq V \backslash K$, where $K := I \triangle J$*

$$p_I(y|w, k) = p_J(y|w, k) \text{ if } Y \perp\!\!\!\perp K | W \text{ in } D_{\underline{K}}. \tag{5}$$

Rule 3 (do-do): *For any $I, J \subseteq V$ and disjoint $Y, W \subseteq V \backslash K$, where $K := I \triangle J$*

$$p_I(y|w) = p_J(y|w) \text{ if } Y \perp\!\!\!\perp K | W \text{ in } D_{\overline{K(W)}}. \tag{6}$$

Rule 4 (mixed do-do/do-see): *For any $I, J \subseteq V$ and disjoint $Y, W \subseteq V$, where $K := I \triangle J$*

$$p_I(y|w) = p_J(y|w) \text{ if } Y \perp\!\!\!\perp K | W \backslash W_k \text{ in } D_{\underline{W_k}, \overline{R(W)}}, \tag{7}$$

*where $W_k =: W \cap K$ and $R := K \backslash W_k$.*

Note that Rule 2 and Rule 3 are special cases of Rule 4. We present all three to make the connection to standard causal calculus rules more explicit.

*Proof.* Let $K_I := I \backslash J, K_J := J \backslash I, T := I \cap J$.

**Rule 1**: The result follows from the rule 1 of Theorem 1.

**Rule 2**: We have the following lemma:

**Lemma 1.** *If $Y \perp\!\!\!\perp K | W$ in $G_{\underline{K}}$ then $Y \perp\!\!\!\perp K_I | W, K_J$ in $G_{\underline{K_I}}$ and $Y \perp\!\!\!\perp K_J | W, K_I$ in $G_{\underline{K_J}}$.*

*Proof.* Suppose for the sake of contradiction that $Y \perp\!\!\!\perp K_I | W, K_J$ in $G_{\underline{K_I}}$ does not hold, then there exist a corresponding active path, denoted $p$. If every collider along $p$ is active due to a node in $W$ and not $K_J$, then $p$ is active in $G_{\underline{K}}$ as well which contradicts the input. Otherwise, let $K_J^* \in K_J$ be the node activating the last collider $S$ along $p$ (where possibly $K_J^* = S$) starting from $K_I$. The path $p'$ composed of the directed path from $S$ to $K_J^*$ concatenated with the subpath of $p$ from $S$ to $Y$ is active in $G_{\underline{K}}$ which contradicts the input. Hence, $Y \perp\!\!\!\perp K_I | W, K_J$ in $G_{\underline{K_I}}$. Similarly, we can show that $Y \perp\!\!\!\perp K_J | W, K_I$ in $G_{\underline{K_J}}$. □

Therefore we can apply rule 2 of Theorem 1 to obtain $p_I(y|w, k) = p_T(y|w, k)$. Furthermore, we can apply rule 2 of Theorem 1 once more to obtain $p_T(y|w, k) = p_J(y|w, k)$, which concludes the proof.

**Rule 3**: We have the following lemma:

**Lemma 2.** *If $Y \perp\!\!\!\perp K | W$ in $G_{\overline{K(W)}}$, then $Y \perp\!\!\!\perp K_I | W$ in $G_{\overline{K_I(W)}}$ and $Y \perp\!\!\!\perp K_J | W$ in $G_{\overline{K_J(W)}}$.*

*Proof.* If $Y \perp\!\!\!\perp K | W$ in $G_{\overline{K(W)}}$, then clearly $Y \perp\!\!\!\perp K_I | W$ in $G_{\overline{K(W)}}$. Suppose for the sake of contradiction, we have $Y \not\perp\!\!\!\perp K_I | W$ in $G_{\overline{K_I(W)}}$. Notice that the only difference between $G_{\overline{K(W)}}$ and $G_{\overline{K_I(W)}}$ are the incoming edges into $K_J(W)$. Therefore, the active path $p$ between $K_I$ and $Y$ in $G_{\overline{K_I(W)}}$ must include a vertex $S \in K_J(W)$ and also must pass through an edge that is into $S$. Otherwise, $p$ would be active in the graph $G_{\overline{K(W)}}$ which contradicts the input. Since no descendant of $K_J(W)$ is conditioned on by definition, no descendant of $S$ is conditioned on. Also, since $p$ is active, then $S$ cannot be a collider on $p$. This implies that the other edge that is adjacent to $S$ must be out of it. Moreover, along the subpath of $p$ that is out of $S$, denoted $p'$, none of the nodes is a collider. Suppose otherwise for the sake of contradiction and let $X$ be the first collider. since $p$ is active, then we condition on a

descendant of $X$. Since the path from $S$ to $X$ is a directed path out of $S$, this contradicts the condition that $S$ is not an ancestor of a node in $W$. Therefore, $p'$ is a directed path out of $K_J$ and can be either into $K_I$ or into $Y$. If $p'$ is into $A \in K_I$, then it must be that $A \notin K_I(W)$. So, $S$ is an ancestor of $A$ and $A$ is an ancestor of some node in $W$ which contradicts the condition $S \in K_J(W)$. Hence, $p'$ must be a directed path out of $S$ and into $Y$. This path is active in $G_{\overline{K_I(W)}}$ and consequently in $G_{\overline{K(W)}}$ which contradicts the separation statement in the assumption. Hence, $Y \perp\!\!\!\perp K_I | W$ in $G_{\overline{K_I(W)}}$. Similarly, we can show that $Y \perp\!\!\!\perp K_J | W$ in $G_{\overline{K_J(W)}}$. $\qquad\square$

Since, $Y \perp\!\!\!\perp K_I | W$ in $G_{\overline{K_I(W)}}$, then we have $p_I(y|w) = p_T(y|w)$ by rule 3 of Theorem 1. Similarly, since $Y \perp\!\!\!\perp K_J | W$ in $G_{\overline{K_J(W)}}$ we have $p_T(y|w) = p_J(y|w)$. This concludes the proof.

**Rule 4**: In addition to the notation defined in rule 4, let $W_I := W_k \cap I$, $W_J := W_k \cap J$, $R_I := R \cap I$, $R_J := R \cap J$. The following venn diagram summarizes those relations.

First, we establish the following. Note that $R_I \cup R_J = R$ and $W_I \cup W_J = W_k$.

**Lemma 3.** *If $Y \perp\!\!\!\perp K | W \setminus W_k$ in $D_{\underline{W_k}, \overline{R(W)}}$, then $Y \perp\!\!\!\perp R | W$ in $D_{\overline{R(W)}}$ and $Y \perp\!\!\!\perp W_k | W \setminus W_k$ in $D_{\underline{W_k}}$.*

*Proof.* If $Y \perp\!\!\!\perp K | W \setminus W_k$ in $D_{\underline{W_k}, \overline{R(W)}}$, then $Y \perp\!\!\!\perp R | W \setminus W_k$ in $D_{\underline{W_k}, \overline{R(W)}}$ since $R \subset K$. Suppose for the sake of contradiction that $Y \not\perp\!\!\!\perp R | W$ in $D_{\overline{R(W)}}$ and let $p$ be one active path between $Y$ and $R$. The difference between $D_{\underline{W_k}, \overline{R(W)}}$ and $D_{\overline{R(W)}}$ is cutting the edges out of $W_k$. Hence, $p$ is discontinued or blocked in $D_{\underline{W_k}, \overline{R(W)}}$ conditioned on $W \setminus W_k$ due to one of two conditions: (1) $p$ includes a non-collider node in $W_k$, or (2) $p$ has a collider $S$ that is active because it has a descendant in $W_k$ (possibly $S \in W_k$). Case (1) is not possible because $W_k \subset W$ and $p$ would be blocked in $D_{\overline{R(W)}}$ which contradicts the assumption that $p$ is active. Consider the collider along $p$ closest to $Y$ that is consistent with case (2). The directed path from $S$ to the node in $W_k$ concatenated with the subpath of $p$ from $S$ to $Y$ is active given $W \setminus W_k$ in $D_{\underline{W_k}, \overline{R(W)}}$ which contradicts the input condition. Thus, $Y \perp\!\!\!\perp R | W$ in $D_{\overline{R(W)}}$ and this concludes the proof of first part.

If $Y \perp\!\!\!\perp K | W \setminus W_k$ in $D_{\underline{W_k}, \overline{R(W)}}$, then $Y \perp\!\!\!\perp W_k | W \setminus W_k$ in $D_{\underline{W_k}, \overline{R(W)}}$ since $W_k \subset K$. Suppose for the sake of contradiction that $Y \not\perp\!\!\!\perp W_k | W \setminus W_k$ in $D_{\underline{W_k}}$ and let $p$ denote any active path. The only difference between the two graphs is the set of incoming edges to $R(W)$. Therefore, $p$ contains an edge into a node $S \in R(W)$ so that $p$ is active in $D_{\underline{W_k}}$ and blocked in $D_{\underline{W_k}, \overline{R(W)}}$. Since $R(W)$ are by definition non-ancestors of $W$, $S$ cannot be a collider in $D_{\underline{W_k}}$ otherwise it would be blocked. Since $S$ is a non-collider, the other edge adjacent to $S$ must be out of $S$. Moreover, along the subpath of $p$ that is out of $S$, denoted $p'$, none of the nodes is a collider. Suppose otherwise for the sake of contradiction and let $X$ be the first collider. since $p$ is active, then we condition on a descendant of $X$. Since the path from $S$ to $X$ is a directed path out of $S$, this contradicts the condition that $S$ is not an ancestor of a node in $W$ ($S \in R(W)$). Therefore, $p'$ is a directed path out of $S$ and can be either into $Y$ or into a node in $W_k$. It $p'$ is into $W_k$, then $S$ is an ancestor of a node in $W$ which is a contradiction since $S \in R(W)$. If $p'$ is into $Y$ then $p'$ is active in $D_{\underline{W_k}, \overline{R(W)}}$ which contradicts the input condition that $Y \perp\!\!\!\perp K | W \setminus W_k$. This concludes the proof of the second claim. $\qquad\square$

We establish the following equivalences which prove rule 4. Note that the first and the last equivalences follows by definition.

$$p_I(y|w) = p_{R_I \cup W_I \cup T}(y|w) = p_{R_J \cup W_I \cup T}(y|w) = p_{R_J \cup W_J \cup T}(y|w) = p_J(y|w)$$

The second equality is an application of rule 3 since $Y \perp\!\!\!\perp R | W$ in $D_{\overline{R(W)}}$ and the third equality is an application of rule 2 since $Y \perp\!\!\!\perp W_k | W \setminus W_k$ in $D_{\underline{W_k}}$. This concludes the proof. $\qquad\square$

*Proof of Corollary 1.* The correctness follows by Rule 4 of Proposition 2. $\qquad\square$

We have the following lemma which plays an important role in the proof of our graphical characterization of the equivalence class. The proof can be found within the proof of Theorem 2.

**Lemma 4.** *Consider a causal graph with latent variables where either of the graphical conditions in Rules 1,2,3,4 does not hold. Then there exists a tuple of interventional distributions $(p_I)_{I \in \mathcal{I}}$ that is $\mathcal{I}$-Markov to D and the corresponding invariance relation does not hold.*

In other words, the lemma above shows that the causal calculus rules are tight: For graphs where the graph separation statement does not hold, one can obtain interventional distributions where the corresponding invariance fails.

## 7.3 Generalized do-calculus Graph Mutilations and F-node Equivalence

We show graphical conditions on the augmented graph that are equivalent to those given in the generalized causal calculus rules.

**Proposition 3.** *Consider a CBN ($D = (V \cup L, E)$, p) with latent variables L and its augmented graph $Aug_{\mathcal{I}}(D) = (V \cup L \cup \mathcal{F}, E \cup \mathcal{E})$ with respect to an intervention set $\mathcal{I}$, where $\mathcal{F} = \{F_i\}_{i \in [k]}$. Let $S_i$ be the set of nodes adjacent to $F_i, \forall i \in [k]$. We have the following equivalence relations:*

*Suppose Y, Z, W are disjoint subsets of V. We have*

$$(Y \perp\!\!\!\perp Z | W)_D \iff (Y \perp\!\!\!\perp Z | W, F_{[k]})_{Aug(D)} \tag{8}$$

*For each $S_i$, suppose Y, W are disjoint subsets of $V \setminus S_i$. We have*

$$(Y \perp\!\!\!\perp S_i | W)_{D_{\underline{S_i}}} \iff (Y \perp\!\!\!\perp F_i | W, S_i, F_{[k]\setminus\{i\}})_{Aug(D)} \tag{9}$$

$$(Y \perp\!\!\!\perp S_i | W)_{D_{\overline{S_i(W)}}} \iff (Y \perp\!\!\!\perp F_i | W, F_{[k]\setminus\{i\}})_{Aug(D)} \tag{10}$$

*For each $S_i$, let $Y \subseteq V$ and $W \subseteq V$. Let $W_i := W \cap S_i, R := S_i \setminus W_i$. Then we have*

$$(Y \perp\!\!\!\perp S_i | W \setminus W_i)_{D_{\underline{W_i}, \overline{R(W)}}} \iff (Y \perp\!\!\!\perp F_i | W, F_{[k]\setminus\{i\}})_{Aug(D)} \tag{11}$$

*Proof.* Conditioning on a source node is equivalent to removing it from the graph in terms of the graph separation statements. Hence, conditioning on $F_{[k]\setminus\{i\}}$ in the right-hand side eliminates them. Therefore, equations (8), (9), and (10) follow from [18, Proof of Th. 4.1] by Pearl. In what follows, we prove (11).

We first consider the case when $Y \cap S_i \neq \emptyset$. Then the relation is trivially true since it implies that for some $U \in S_i$, U and $F_i$ are adjacent in $Aug(D)$ and Y is dependent with U since $U \subseteq Y$.

In the rest of the proof, suppose $Y \subseteq V \setminus S_i$.

Suppose $(Y \not\perp\!\!\!\perp S_i | W \setminus W_i)_{D_{\underline{W_i}, \overline{R(W)}}}$, and let p denote any active path from $A \in S_i$ to Y. Note that the same path is active in $Aug(D)$ given $W, F_{[k]\setminus\{i\}}$. If p is into A, then either (1) $A \in W_i$ or (2) $A \notin R(W)$. Hence, the concatenation of p with $F_i \to A$ is active in $Aug(D)$ given $W, F_{[k]\setminus\{i\}}$ since $A \in W$ for case (1) and A has a descendant in W for case(2). Hence, $(Y \not\perp\!\!\!\perp F_i | W, F_{[k]\setminus\{i\}})_{Aug(D)}$.

Next, suppose $(Y \not\perp\!\!\!\perp F_i | W, F_{[k]\setminus\{i\}})_{Aug(D)}$ and let p denote any active path. Also, let A be the closest node to Y along p such that A is active due to a node in $S_i$, i.e., $A \in S_i$ is along p or $A \notin S_i$ is an active collider due to a descendant in $W_i \subseteq S_i$. If A is a non-collider along p, then $A \in R \subseteq S_i$ else p is blocked. If the subpath from A to Y is out of A, then this subpath is active in $D_{\underline{W_i}, \overline{R(W)}}$ given $W \setminus W_i$ and $(Y \not\perp\!\!\!\perp S_i | W \setminus W_i)_{D_{\underline{W_i}, \overline{R(W)}}}$. Otherwise, the subpath between A and $F_i$ is out of A. In this case, we argue that $A \notin R(W)$, hence the subpath from A to Y along p is active in $D_{\underline{W_i}, \overline{R(W)}}$ given $W \setminus W_i$ and $(Y \not\perp\!\!\!\perp S_i | W \setminus W_i)_{D_{\underline{W_i}, \overline{R(W)}}}$. Since all the edges incident on $F_i$ are out of it, then there exist at least one collider between A and $F_i$ along p. Let X denote such a collider closest to

*A*. Since *X* is active, then *X* has a descendant in *W*, thus *A* has a descendant in *W* through *X* and $A \notin R(W)$. Alternatively, *A* is an active collider along *p*. If $A \in W_i$ or $A \notin R(W)$, then the path from *A* to *Y* is active and $(Y \not\perp\!\!\!\perp S_i | W \setminus W_i)_{D_{W_i, \overline{R(W)}}}$. Not that *A* can't be in $R(W)$, else *A* would be blocked along *p*. Finally, $A \notin S_i$ and it has a descendant in $W_i$. In this case, the directed path from *A* to the node in $W_i$ concatenated with the subpath of *p* from *A* to *Y* is active in $D_{\underline{W_i}, \overline{R(W)}}$ given $W \setminus W_i$ and $(Y \not\perp\!\!\!\perp S_i | W \setminus W_i)_{D_{\underline{W_i}, \overline{R(W)}}}$. This concludes the proof. □

*Proof of Proposition 1.* This follows from Proposition 3. □

## 7.4 Proof of Theorem 2

Suppose that $MAG(Aug_{\mathcal{I}}(D_1))$ and $MAG(Aug_{\mathcal{I}}(D_2))$ satisfy the three conditions. Then, they induce the same m-separation statements and vice-versa [26, Prop. 1 & Def. 5]. It follows by Proposition 1 that $\mathcal{D}_1$ and $\mathcal{D}_2$ impose the same constraints over the distribution tuples in Definition 1. Therefore, $\mathcal{P}_{\mathcal{I}}(D_1, V) = \mathcal{P}_{\mathcal{I}}(D_2, V)$.

For the other direction, suppose $MAG(Aug_{\mathcal{I}}(D_1))$ and $MAG(Aug_{\mathcal{I}}(D_2))$ do not satisfy the three conditions. Then, they must induce at least one different m-separation statement. Therefore, we need to establish that if the two graphs induce different m-separation statements, then they are not $\mathcal{I}$-Markov equivalent.

Before we show the other direction, we need to introduce some definitions and establish some results.

Define the following collections of m-separation statements on the $Aug(D)$:

$$\mathcal{U} = \{(X \perp\!\!\!\perp Y | Z, F)_{Aug(D)} : X, Y \in V \cup \mathcal{F}, \; Z \subseteq V - \{X, Y\}, \; F \subsetneq \mathcal{F} - \{X, Y\}.\} \tag{12}$$

$$O = \{(X \perp\!\!\!\perp Y | Z, F)_{Aug(D)} : X, Y \in V \cup \mathcal{F}, \; Z \subseteq V - \{X, Y\}, \; F = \mathcal{F} - \{X, Y\}.\} \tag{13}$$

$$\mathcal{T} = \{(X \perp\!\!\!\perp Y | Z, F)_{Aug(D)} : X \in V, \; Y \in V \cup \mathcal{F}, \; Z \subseteq V - \{X, Y\}, \; F = \mathcal{F} - \{X, Y\}.\} \tag{14}$$

$\mathcal{U}$ are the set of m-separation statements between any two nodes given a strict subset of all the remaining F nodes. $O$ are the set of m-separation statements between any two nodes given all the remaining F nodes. $\mathcal{T}$ are the set of m-separation statements between an observable node and any other node given all the remaining F nodes. Note that $\mathcal{U}, O$ are disjoint, whereas $\mathcal{T}$ is a subset of $O$. From Prop. 1 and Def. 1, we see that an m-separation statement is in $\mathcal{T}$ if and only if it appears as a graphical condition in the definition of $\mathcal{I}$-Markov equivalence class of distributions for *D*. Also, if an m-separation between arbitrary subsets of nodes holds in $D_1$ but not in $D_2$, then there is at least one pair of singletons for which the corresponding m-separation holds in $D_1$ but not in $D_2$. Therefore it is sufficient to consider m-separation statements between singletons which are included in $\mathcal{U} \cup O \cup \mathcal{T}$.

**Lemma 5.** *Suppose* $(A \perp\!\!\!\perp B | C)_{Aug(D_1)}, (A \not\perp\!\!\!\perp B | C)_{Aug(D_2)}$*, where A, B, C are arbitrary disjoint subsets of* $V \cup \{F_{[k]}\}$*. Then at least one of the following is true:*

*(a)* $\exists X, Y, Z \subseteq V$ *such that* $(X \perp\!\!\!\perp Y | Z, \mathcal{F})_{Aug(D_1)}$ *AND* $(X \not\perp\!\!\!\perp Y | Z, \mathcal{F})_{Aug(D_2)}$ $\tag{15}$

*(b)* $\exists T, W \subseteq V, F_i \in \mathcal{F}$ *such that* $(F_i \perp\!\!\!\perp T | W, \mathcal{F} - \{F_i\})_{Aug(D_1)}$ *AND* $(F_i \not\perp\!\!\!\perp T | W, \mathcal{F} - \{F_i\})_{Aug(D_2)}$ $\tag{16}$

*Proof Sketch.* The statement of the lemma can be rephrased as follows: Any difference in the truth value of any m-separation statement from the set $\mathcal{U} \cup O \cup \mathcal{T}$ between $Aug(D_1)$ and $Aug(D_2)$ implies a difference between truth value of some m-separation statement in $\mathcal{T}$ between $Aug(D_1)$ and $Aug(D_2)$. We show this in two steps:

1. For any $Aug(D)$, any m-separation statement in $\mathcal{U}$ can be written as a deterministic function of the m-separation statements in $O$. Further, this deterministic function does not depend on the structure of *D*. Therefore, any difference in the truth value of any m-separation statement from the set $\mathcal{U} \cup O \cup \mathcal{T}$ between $Aug(D_1)$ and $Aug(D_2)$ implies a difference between the truth values of some m-separation statement in $O$ between $Aug(D_1)$ and $Aug(D_2)$.

2. If there is any difference in truth value of any m-separation statement in $O$ between $Aug(D_1)$ and $Aug(D_2)$, then this implies a difference in the truth value of some m-separation statement in $\mathcal{T}$ between the augmented graphs.

□

*Detailed Proof of Lemma 5.* We show proof of both the steps outlined in the proof sketch of the Lemma.

**Proof of Step 1:**

The main result in this step is given by Corollary 2. We have the following Lemma that relates m-separation statements from $\mathcal{U}$ to other m-separation statements that are 'closer' to $\mathcal{O}$. Recursively applying this lemma proves the result in this step.

**Lemma 6.** *Let Aug(D) be the augmented graph (augmented with variables in $\mathcal{F}$) with respect to a CBN with latents $(D, p)$. Consider an m-separation statement with respect to Aug(D) of the form $(X \perp\!\!\!\perp Y | Z, F_S )_{Aug(D)}$ where $X, Y \in V \cup \mathcal{F}$ and $Z \subseteq V - \{X, Y\}$ and $F_S \subsetneq \mathcal{F} - \{X, Y\}$. For any $F_i \in \mathcal{F} - (F_S \cup \{X\} \cup \{Y\})$, the following statements are equivalent*

$$(a)\ (X \perp\!\!\!\perp Y | Z, F_S )_{Aug(D)} \tag{17}$$

$$(b)\ (X \perp\!\!\!\perp Y | Z, F_S \cup \{F_i\})_{Aug(D)} \text{AND}[(F_i \perp\!\!\!\perp Y | Z, F_S )_{Aug(D)} \text{OR} (F_i \perp\!\!\!\perp X | Z, F_S )_{Aug(D)}] \tag{18}$$

*Proof.* From the hypothesis in the lemma, $X, Y \neq F_i$ and $F_i \notin F_S$. Suppose there is a m-connecting path between $X$ and $Y$ given $Z, F_S$. Then either it does not pass through $F_i$, which implies $(X \not\perp\!\!\!\perp Y | Z, F_S \cup \{F_i\})_{Aug(D)}$ or it can be decomposed into two paths, one m-connecting $F_i$ and $Y$ given $Z, F_S$ and another m-connecting $F_i$ and $X$ given $Z, F_S$. Note that this is because all arrows are out of $F_i$ by construction of $Aug(D)$ and $F_i$ is not part of the conditioning set. On the other hand, if there is no m-connecting path between $X$ and $Y$ given $Z, F_S$ all the aforementioned paths has to be m-separating which gives the equivalence. □

**Remark:** Please note that Lemma 6 does not depend on the structure of $D$. Accordingly, we have the following corollary:

**Corollary 2.** *Any m-separation statement $X \perp\!\!\!\perp Y | Z, F_S \in \mathcal{U}$ can be written as a deterministic function of the m-separation statements in $\mathcal{O}$. This function is independent of the structure of $D$.*

*Proof.* We keep repeatedly applying (18) until all the formulas begin to lie in $\mathcal{O}$. In each of the expansions using (18), either an unconditioned $F_i$ is added to the conditioning set or it appears as a new conditional independence statement between $F_i$ and $X$ and $Y$ given the current conditioning set. □

**Proof of Step 2:** We only need to focus on the m-separation statements in $\mathcal{O}$ that are not in $\mathcal{T}$. Those are precisely the m-separation statements between two F-nodes given a subset of the observed variables and all the other F-nodes. Suppose in $Aug(D_1)$, $F_i \perp\!\!\!\perp F_j | W, \mathcal{F} - \{i, j\}$ and in $Aug(D_2)$ $F_i \not\perp\!\!\!\perp F_j | W, \mathcal{F} - \{i, j\}$ for some $W \subset V$. Since $F$-nodes are source nodes, the active path between $F_i$ and $F_j$ must contain at least one collider. Consider the shortest path that is active in $Aug(D_2)$ but not in $Aug(D_1)$. Suppose the active path between $F_i$ and $F_j$ contains a single collider. This can only happen if in $Aug(D_2)$, $\exists t \in W$ s.t. $t \in De(F_i) \cap De(F_j)$, otherwise no descendant of any collider on the path would be conditioned on, and in $Aug(D_1)$ $\nexists t \in W$ s.t. $t \in De(F_i) \cap De(F_j)$. This means in $Aug(D_1)$, $t$ is either not a descendant of $F_i$ or it is not a descendant of $F_j$. Suppose without loss of generality, $t$ is not a descendant of $F_i$ in $Aug(D_1)$ but it is in $Aug(D_2)$. This implies that in $Aug(D_1)$, $F_i \perp\!\!\!\perp t | \mathcal{F} - \{i\}$ and in $Aug(D_2)$, $F_i \not\perp\!\!\!\perp t | \mathcal{F} - \{i\}$. This shows that some m-separation statement belonging to $\mathcal{T}$ is different in the two graphs.

Now suppose that the active path between $F_i, F_j$ contain at least two colliders. Consider the collider on the path that is closest to $F_i$, and call this node $T_i$. Similarly, let us call the collider closest to $F_j$ on the active path as $T_j$. $T_i$ and $T_j$ must have descendants that are in $W$ since the path is active. Consider the subpath between $F_i$ and $T_j$ and call this $p_1$. Consider the subpath between $T_i$ and $F_j$ and call this path $p_2$. Note that in $Aug(D_2)$, the union $p_1 \cup p_2$ is active and $p_1, p_2$ are overlapping since colliders are distinct. Since $p$ is active, the subpaths $p_1, p_2$ should also be active in $Aug(D_2)$. Now note that this path is not active in $Aug(D_1)$. This means that either $p_1$ or $p_2$ is not active because otherwise, since $p_1$ and $p_2$ are overlapping, if they were active, their union would be active as well. Therefore either $p_1$ or $p_2$ create different m-separation statements in $Aug(D_1)$ compared to $Aug(D_2)$. Suppose without

loss of generality that $p_1$ is active in $Aug(D_2)$ but not in $Aug(D_1)$. Therefore $(F_i \not\perp\!\!\!\perp T_j | \mathcal{F} - F_i)_{Aug(D_2)}$ and $(F_i \perp\!\!\!\perp T_j | \mathcal{F} - F_i)_{Aug(D_1)}$, both of which are testable statements. This concludes the proof.

We can finally prove Lemma 5. Suppose $(A \perp\!\!\!\perp B | C)_{Aug(D_1)}, (A \not\perp\!\!\!\perp B | C)_{Aug(D_2)}$. Any m-separation statement belongs to one of $\mathcal{O}, \mathcal{U}, \mathcal{T}$. Note also that vertex set of a graph determines which set it belongs to. Therefore the same m-separation statement for $Aug(D_1), Aug(D_2)$ belong to the same set since both have the same vertex set.

*(a)* If it belongs to $\mathcal{T}$, we are done.

*(b)* If it belongs to $\mathcal{O}$, then by Step 2, any m-separation statement with different truth values imply that an m-separation statement has different truth values in $\mathcal{T}$ and result follows from *(a)*.

*(c)* If it belongs to $\mathcal{U}$, then by Step 1, the m-separation statement is a deterministic function of m-separation statements of $\mathcal{O}$. Since m-separation statements in $\mathcal{U}$ have different truth values, at least one of the m-separation statements in $\mathcal{O}$ that determines the original m-separation statement in $\mathcal{U}$ via this function must be different. The result follows from *(b)*. □

We showed that if $MAG(Aug(D_1))$ and $MAG(Aug(D_2))$ are not Markov equivalent, then there is an m-separation statement that appears as a condition in the definition of $\mathcal{I}$-Markov equivalence that is different in the two graphs: There is an m-separating path in $Aug(D_1)$ that is m-connecting in $Aug(D_2)$. In order to complete the proof, we need to show that $\mathcal{P}_{\mathcal{I}}(D_2)$ contains tuples of distributions that are not in $\mathcal{P}_{\mathcal{I}}(D_1)$. This is shown in the following Lemma, which concludes the proof.

**Proof of Lemma 4**:

For this, we leverage a key result of Meek which he used to show that the set of unfaithful distributions has Lebesgue measure zero, combining it with a jointly Gaussian structural causal model construction including the latent variables. We first state Meek's result as a standalone lemma:

**Lemma 7** (Meek). *Consider a causal DAG $D = (V, E)$, where $(A \not\perp\!\!\!\perp B | C)_D$. Let $D_s = (V_s, E_s)$ be the subgraph that contains all the nodes in the m-connecting path that induce $(A \not\perp\!\!\!\perp B | C)_D$. Then any distribution $p$ over $V_s$ where every adjacent pair of variables are dependent satisfies $(A \not\perp\!\!\!\perp B | C)_p$.*

*Proof.* Proof uses weak transitivity and an inductive argument and can be found in [14]. □

Suppose that $X, Y, Z \subseteq V$ such that $(X \perp\!\!\!\perp Y | Z, \mathcal{F})_{Aug(D_1)}$ AND $(X \not\perp\!\!\!\perp Y | Z, \mathcal{F})_{Aug(D_2)}$. Suppose that both $X, Y$ are observed variables. In this case, any tuple of interventional distribution obtained from an observational distribution that is faithful to the causal graph with latent variables constitute a valid example.

Suppose $X = F_i$ for some $i \in [k]$ and $Y \in V$. Therefore, an F-node is m-connected to an observed node in $Aug(D_2)$ but not in $Aug(D_1)$.

Consider the causal graph $D_2 = (V \cup L, E)$ with latents. Focus on the subgraph of $D_2$ that includes all the variables that contribute to the m-connecting path of $(X \not\perp\!\!\!\perp Y | Z, \mathcal{F})_{Aug(D_2)}$. An example is in [14]. Let us call this subgraph $D_{path} = (V_{path}, E_{path})$. Consider a jointly Gaussian distribution on $V_{path}$ that is faithful to $D_{path}$. One exists by construction of Meek (Theorem 7 of [14]). Let us call this distribution $p_{path}$. We will only focus on this distribution only to finally expand it by adding the remaining variables as jointly independent and independent from the variables in $D_{path}$. Consider two interventions $I, J$ on the causal Bayesian network $(D_{path}, p_{path})$, where $I \triangle J = S_i$, i.e., the distributions $p_I, p_J$ are responsible for the graphical separation of $F_i$. Different from the rest of the paper, for this proof we will treat $F_i$ as a regime variable that indicates when we switch to $p_I$ and when we switch to $p_J$. Note that we can do this since we only add this single F node and no others. Consider the distribution $p^*$ defined as follows: $p^*(.|F_i = 0) = p_I(.), p^*(.|F_i = 1) = p_J$. Also pick the uniform distribution for $F_i$. We need to show that the invariances that are implied by the graph separation in question in the generalized causal calculus rules fails for $p_I, p_J$. This is equivalent to showing that *the variable $F_i$ is dependent with $Y$ given $Z$ on the distribution $p^*$*. We construct the interventional distributions through an SCM which implies the CBN in question. This is done by the simply adding extra noise terms to the structural equations describing the CBN.

Let **x** be a vector representing all the variables in the graph including the latents. Consider the following structural equation model: Let $\mathbf{x} = \mathbf{A}\mathbf{x} + \mathbf{e}$, where **A** is the lower triangular matrix that captures the graph structure and parental relations in $D_{path}$ and **e** is the exogenous noise vector. Let

$p_I$ be the distribution obtained by adding the noise vector $\mathbf{e}_J$ to the system. $\mathbf{e_I}$ is non-zero in the rows $i$ if $x_i \in I$. Therefore $p_I$ is a valid soft interventional distribution. Similarly, let $\mathbf{e}_J$ be the noise vector added for intervention on $J$. Next, we show that in the combined distribution $q$ using these $p_I, p_J$ every adjacent variable are dependent. Clearly, when $\mathbf{e}_1$ and $\mathbf{e}_2$ are different, F-variable is dependent with the variables in $K := I \triangle J$, since $p(K|F = 0) \neq p(K|F = 1)$, which implies $(K \not\perp F |\emptyset)_{p^*}$. Therefore, we focus on establishing that every pair of variables that are adjacent are correlated except for the F variable. The correlation of the variables in $D_{path}$ matrix can be calculated as follows:

$$\mathbf{x} = \mathbf{Ax} + \mathbf{e} + \mathbf{e}_I \Rightarrow (\mathbf{I} - \mathbf{A})\mathbf{x} = \mathbf{e}_1 \Rightarrow \mathbf{x} = (\mathbf{I} - \mathbf{A})^{-1}\mathbf{e}_1 \tag{19}$$

$$\mathbf{x} = \mathbf{Ax} + \mathbf{e} + \mathbf{e}_J \Rightarrow (\mathbf{I} - \mathbf{A})\mathbf{x} = \mathbf{e}_2 \Rightarrow \mathbf{x} = (\mathbf{I} - \mathbf{A})^{-1}\mathbf{e}_2 \tag{20}$$

where $\mathbf{e}_1 = \mathbf{e} + \mathbf{e}_I$ and $\mathbf{e}_2 = \mathbf{e} + \mathbf{e}_J$. The correlation matrix between the observed variables with respect to $p^*(.)$ can be calculated as follows (since the binary regime variable will be marginalized out):

$$E[\mathbf{xx}^T] = 0.5(\mathbf{I} - \mathbf{A})^{-1}E[\mathbf{e}_1\mathbf{e}_1^T](\mathbf{I} - \mathbf{A})^{-1^T} + 0.5(\mathbf{I} - \mathbf{A})^{-1}E[\mathbf{e}_2\mathbf{e}_2^T](\mathbf{I} - \mathbf{A})^{-1^T} \tag{21}$$

$$= 0.5(\mathbf{I} - \mathbf{A})^{-1}(\mathbf{D}_1 + \mathbf{D}_2)(\mathbf{I} - \mathbf{A})^{-1^T}, \tag{22}$$

where $\mathbf{D}_1 = E[\mathbf{e}_1\mathbf{e}_1^T]$ and $\mathbf{D}_2 = E[\mathbf{e}_2\mathbf{e}_2^T]$ are diagonal covariance matrices of the noise added via soft interventions. Consider two adjacent variables $x_i, x_j$ in $D_{path}$. We have a few observations: $\mathbf{I} - \mathbf{A}$ is a full rank matrix since $\mathbf{A}$ is a strictly lower triangular matrix, hence it's inverse exists and is unique. We treat $\mathbf{D}_1$ and $\mathbf{D}_2$ as variables in this system: When we perform the soft intervention, we get to choose the variance of each added noise term. We want to show that there always exist soft interventions, i.e., $\mathbf{D}_1, \mathbf{D}_2$ such that $x_i, x_j$ are dependent. Since $x_i, x_j$ are jointly Gaussian, they are dependent if and only if they are correlated. Hence, we only need to show that $E[x_i x_j] \neq 0$ for any adjacent pair $x_i, x_j$. Notice that this condition is equivalent to a linear equation being zero. Therefore, $E[x_i x_j] = 0$ for all $\mathbf{D}_1, \mathbf{D}_2$ or it is non-zero except for a particular value of $\mathbf{D}_1, \mathbf{D}_2$. If we set $\mathbf{D}_1 = \mathbf{D}_2 = \mathbf{0}$, we get back the observational system. By assumption any pair of adjacent variables are dependent since the original distribution is chosen to be faithful to the graph $D_{path}$. Therefore, this system of linear equations is not identically zero. Hence, if we randomly pick the variances of the added noise terms, with probability 1, any adjacent pair of variables will be dependent (after a union bound).

Therefore, we have established that in the graph $D_{path}$ plus the F-variable, every pair of adjacent variables are dependent. Now, we can use Meek's lemma, which gives us that $(F_i \not\perp Y |Z)_q$ (Since we did not add the other F variables as regime variables, we do not need to condition on them.). Now, we can augment this distribution to cover the variables outside $D_{path}$: Simply pick all the remaining variables jointly independent and independent from the variables in $D_{path}$. Construct the interventional distributions by similar soft intervention of adding extra noise terms to the intervened variables. The corresponding tuple of interventional distributions belong to $\mathcal{P}_{\mathcal{I}}(D_2, V)$ but not to $\mathcal{P}_{\mathcal{I}}(D_1, V)$ since m-separation should have implied invariance between the interventional distributions whereas we constructed the interventional distributions such that this is not true. □

## 7.5 Proof of Theorem 3

The main idea of the algorithm is to infer the separating sets between pairs of nodes using the invariance tests. Using c-faithfulness assumption, it is easy to see that the invariances that are checked imply m-separation statements between the nodes of the augmented graph. However, the separating sets that are found always include all the F-nodes. There are few questions we need to address to prove soundness of the algorithm:

*(1) Are all pairs of separable nodes in Aug(D) correctly identified by the algorithm?*

*(2) Does the choice of separating set affect the application of FCI rules.*

*(3) Are the orientation rules sound?*

We first address *(1)*: Note that all pairs of F-nodes are separable with the empty set by construction of $Aug(D)$. This is captured in Line 8 of the algorithm by setting $SepSet(F_i, F_j) = \emptyset$ for all pairs of F-nodes. This assures that after Phase I, they become non-adjacent.

Next consider all pairs $X, Y$ where at least one is not an F-node. Suppose two nodes are separable in $Aug(D)$. Then there is a set $W$ that makes them separable. There is no restriction on $W$: It may

or may not have some of the F-nodes. However, since F-nodes of $Aug(D)$ are always source nodes, adding the remaining F-nodes cannot open new paths. Therefore, the set $W \cup \mathcal{F}$ is also a separating set. Formally, we have the following lemma:

**Lemma 8.** *For any pair $X, Y \in V \cup \mathcal{F}$, if $(X \perp\!\!\!\perp Y | W)_{Aug(D)}$, then $(X \perp\!\!\!\perp Y | W \cup (\mathcal{F} - X \cup Y))_{Aug(D)}$.*

*Proof.* Proof follows from the fact that F-nodes are source nodes in $Aug(D)$ and the rules of m-separation. $\square$

Therefore, any separable pair imply a testable separation statement by Theorem 1 and it will be identified by the algorithm. This addresses *(1)*. We next address *(2)*.

We make use of the following simple observation: Although there may be more than one separating set for a pair of variables in the graph, FCI algorithm is sound and complete irrespective of which separating set is chosen. From the phrasing of the algorithm and its soundness, this is obvious since which separating set should be used is not specified. Here, we verify this by checking how the rules that require use of separating sets are affected by our choice of separating set:

**Orienting unshielded colliders:** Suppose we consider an ordered triple $\langle X, Y, Z \rangle$ where $X, Z$ are non-adjacent. An F-node can never be a collider. Then the only case where the application of the rule may be affected by which separating system is used is when $X, Z \in V, Y \in \mathcal{F}$. Since by construction of $SepSet$, $Y \in SepSet(X, Z)$ algorithm does not orient it as a collider, which is correct. No collider will be missed by the algorithm due to the choice of $SepSet$.

**Discriminating paths:** By definition of discriminating path [28] and construction of augmented graph, there cannot be discriminating paths between pairs of F-nodes. We can have discriminating paths between an F-node and an observed node as $\langle X, \ldots, W, U, Y \rangle$, where $X \in \mathcal{F}$ and $Y \in V$. First, no F-node can be between $X$ and $U$ since by definition of discriminating path, they should be colliders. If $U$ is not an F-node, then the change in separating system, i.e., adding extra F-nodes does not affect how the rule is applied. Suppose $U$ is an F-node. Then by construction of the separating set, it has to be in the separating set. Then the rule is applied to orient $U \rightarrow Y$, which is consistent with Rule 8 and the augmented graph construction.

Finally, we address the soundness of orientation rules to address *(3)*. The rules of FCI are sound as shown by [27]. This is applicable in our setting, as one can see the augmented graph as a CBN with latents, ignoring how F-nodes are constructed, since m-separation statements implied by this CBN, which are purely graph theoretic criteria, are identical to those implied by the augmented graph. Moreover, previous phases of our algorithm are shown to be sound and complete, which is required for the soundness of this step: Skeleton is correctly identified. Moreover, if there is an unshielded collider, previous phases will correctly identify it. This is necessary for the correctness of the orientation rules of FCI. Therefore, we only need to check the soundness of the additional rules Rule 8,9. Soundness of Rule 8 is trivial since in any augmented graph $Aug(D)$, F-nodes are source nodes.

**Soundness of Rule 9:** Consider a pair $F_i, Y$ where $F_i \in \mathcal{F}, Y \in V$ that are adjacent and $Y \ni nS_i$. This means there is no separating set for $F_i, Y$ in $Aug(D)$, although by construction, they are not adjacent. This can only happen if there is an inducing path between $F_i$ and $Y$ relative to the latent variables $L$. An inducing path relative to latents $L$ is defined as follows [28]: A path $l$ in $Aug(D)$ is an inducing path if *i)* every non-endpoint that is not in $L$ is a collider and *ii)* every collider is an ancestor of either endpoints. Since F-nodes by construction do not have ancestors, every collider on the inducing path between $F_i, Y$ must be an ancestor of $Y$. Therefore in $MAG(Aug(D))$, the observed node must be an ancestor of $Y$. If $|S_k| = 1$, then any inducing path must go through the node in $S_i$, since in $Aug(D)$, $F_i$ is only adjacent to the node in $S_i$. Since this node is on an inducing path, it must be an ancestor of $Y$. Therefore $MAG(Aug(D))$ contains an edge from this node to $Y$. This concludes the proof. $\square$