[Reviews · NeurIPS 2019]

Reviewer 1



- Originality: the contributions are original and provide new insights. - Quality: one can discuss what way is the best to approach the problem, but the road taken seems sound. Details below. - Clarity: the most interesting parts are in my opinion not described well enough. I have a feeling that one could provide simple and intuitive explanations of the main idea which would improve readability. - Significance: the problems considered are relevant and the solutions could prove useful. The present paper seems to be a step towards such solutions. Detailed comments: The paper works with soft interventions only, and at the end of the paper the case of hard interventions is mentioned. It says that hard interventions can change the skeleton of the graph, and this is not handled by the present paper. Is it correct that the soft interventions considered should always have the same set of parents in the interventional graph as in the original? This would seem very restrictive. I'm curious about the choice of using MAGs to represent the interventional distributions. The abstract goal of the paper is to describe equivalence classes of graphs that are more fine-grained than the standard ones, in this case by using interventional equivalence as well. The abstract goal seems to be the same as that of nested Markov models which are also more fine-grained models in which constraints induced by interventional distributions are employed (though all of these constraints are necessarily identifiable from the observational distribution only). I think the paper would benefit from including a comparison between the proposed methods and the nested Markov models. This leads to another question. In the nested Markov models, ADMGs (not MAGs) are used as graphical representations of partially observed causal systems. I believe this choice was made because the MAGs throw away some causal information by insisting on using at most one edge between each pair of nodes. To give an example, we can consider the graphs in Figure 2. In the MAG (even without the auxiliary F_x node) there is directed edge from X to Y (because X and Y cannot be separated, and X is an ancestor of Y). This means that when looking at the MAG, it would seem that when intervening on (X,W), the value of X actually matters for the distribution of Y. However, knowing the underlying causal graph in (a), this is not the case. It could indicate that MAGs are actually throwing interesting information away (they of course encode the independence structure, but not necessarily that of the interventional distributions). You could also formulate this as saying that even if an algorithm learns the I-PAG completely, this doesn't provide complete causal information. To take the example in Figure 2, if I understand correctly, we have access to the interventional distribution from intervening on \{X\} and this shows that X is not a parent of Y (condition on W), however, this information would never be in the output when learning equivalence classes of MAGs. The learning algorithm is described very briefly (I acknowledge that the page limit probably also is to blame for this), and this makes the paper unnecessarily difficult to read. As far as I can understand, the interesting part happens in Algorithm 3, Lines 7-11, and I would suggest describing this in more depth. If I understand correctly, we are basically testing separation between intervention nodes and non-intervention nodes by testing whether there is some set (of non-intervention nodes) such that the conditional distribution of Y remains the same between different intervention regimes. This seems like a simple idea, and one worth communicating early in the paper. At the end, the reader could be left wondering how much more information is gained in the algorithm. More information is definitely required as input to the algorithm, as we also need interventional distributions. The example clearly shows that it is possible to obtain more information. If Rule 9 is the only real gain of the algorithm, it would seem that we could just employ that one directly on the output of the FCI algorithm using the observational distribution. Again, if true, this could offer a more intuitive explanation of the crux of the matter. I'm not sure I understand Definition 5 (Augmented MAG). There seems to be no requirement that the augmented MAG is maximal (i.e. unseparable nodes are adjacent) which is different from MAGs. Is this an omission? In Line 248, it says that something was established, but that seems to be by definition. Minor comments: In Line 118-120, some form of faithfulness seems to be assumed. In Line 100 it says that randomized controlled trials are also called interventions which I don't think is true. Some times people say that interventions can be thought of as generalized randomized controlled trials. I'm not sure what you mean by that sentence. In Lines 118-127, I would suggest adding some references, e.g. to the paper showing completeness of FCI, etc. In Definition 2, the notation [k] has not been introduced. In Lines 200-201, 'since augmented' -> 'since the augmented' Algorithm 2: 2^V is the power set of V? In line 9 in Algorithm 3, it looks like I, J is set to the same thing, maybe write 'I,J is s.t. (I,J) = \sigma(i)'. I think this would be more easily understood. EDIT after response from authors: I thank the authors for providing answers to my questions and concerns. This clarifies central parts of the paper in my view and I've updated my score to reflect this.

Reviewer 2



Update: the author has successfully addressed my concerns. Original comments: In this paper, the authors proposed a new characterization of the interventional equivalence class for learning DAGs with latent variables from soft interventional data. The authors also introduced a new algorithm based on such new characterization. Pros. 1. The paper is well written. In introduction and preliminaries the authors gave a complete overview of previous work in this area, including basic Pearl’s do-calculus, characterization of equivalence in the causal sufficient setting as well as causal inference in the latent variable setting; 2. I agree that the problem considered in this paper is very important. So far a large body of methods have been proposed to perform causal inference with interventional data in the latent variable setting. However, the characterization of the degree of identifiability seems still missing. This paper presents an important work that filled this gap; 3. I did not check the proof in detail, but the proof looks technically sound. Cons. 1. In lines 173-175, the authors claimed that “For simplicity of exposition, in the rest of the paper we assume that all the interventions are controlled, although this is not required for the characterization.” I am a bit confused about this sentence. Does this mean that the controlled experiment assumption is not necessary? Since this assumption is required for example in the proof of Theorem 4 in the appendix, if this assumption is indeed not necessary, how can you prove Theorem 4 without this assumption? 2. This paper may benefit from a simulation. I am curious how the learn algorithm performs on real data.

Reviewer 3



Updates after author response: - Based on author response and comments from other reviewers, I have raised my score to a 7. - Thank you for clarifying rule 9. The violation of ancestrality is what I was missing if W o-> Y is oriented instead of X o-> Y. This violation is not mentioned in the paper itself as part of the reasoning and is probably important to include. - While reviewer 1 mentioned nested Markov models as retaining more information, the author response is insightful, in that this is true only in the case of hard interventions. In addition, the use of nested Markov models at the moment is restricted due to lack of an FCI type procedure, and indeed characterization of nested Markov equivalence, as pointed out in the author response. - The introduction of a new kind of graph, like the augmented MAG here, still requires justification of its edges as being meaningful re: edges from intervention nodes to nodes that are not intervened on. So a short description of this in the supplement would be good. - Again in the supplement, the authors may consider adding an informal analysis of what additional orientation rules may be necessary/not necessary to prove completeness of the proposed algorithm. - Overall, this is a good submission, and I vote for its acceptance. ++++++++++++++++++++++++++++++++++++++++++++ - Does focusing on soft interventions allow us to side-step violations of faithfulness brought about by determinism as discussed in [9]? - Page 2, line 67 do you mean F_X is *not* *m*-separated from Y given X? - What are the advantages of the present work over methods like [9], and [19]? I'm aware that SAT-solver methods may face scalability issues but [9] is also an FCI type method and should be quite fast. Concrete comparisons with simulations would be useful. - In theorem 2, a more elegant characterization of Markov equivalence in MAGs should be preferred. See [a] https://projecteuclid.org/download/pdfview_1/euclid.aos/1247836670. Two MAGs G1 and G2 may be Markov equivalent but a path p may not be discriminating for a vertex U in both G1 and G2. See figure 6 (i) and (ii) in [a]. This means that discriminating paths must be searched for in both graphs and such a search can be super-polynomial in complexity. The preferred characterization is that two MAGs are Markov equivalent if and only if they have the (1) same skeleton and (2) same colliders with order -- theorem 3.7 in [a]. In any case, statements regarding Markov equivalence in MAGs should cite the work in [a]. One could also cite the work in [b] https://link.springer.com/content/pdf/10.1360%2F04ys0023.pdf but this criterion is not used as much. - FCI can only be said to be consistent when the underlying data generating process is Markov and faithful to a MAG. None of the example DGPs shown in this paper correspond to a MAG. Though acyclic directed mixed graphs may be converted to MAGs as described in [23], this seems to present some problems. In a MAG, some directed edges have different interpretations as being visible or invisible (also discussed in [23]). This seems more problematic when discussing directed edges added from intervention nodes in the augmented MAG to nodes other than the variables being intervened on (as in Figure 2 in the paper). What does it mean for such a directed edge to exist between an interventional node that has a very special meaning in this context and a node that it was not meant to intervene on in the original problem formulation. Does the analogue of visible and invisible edges straightforwardly carry over? One could force FCI to rule out the presence of a directed edge between an interventional node and all other variables on the graph besides the nodes being intervened on (in fact the presence of an edge between the interventional node and the intervened node should be enforced as background knowledge) but there is no MAG that is Markov equivalent to figure 2(a) if such edges are disallowed. In light of this, I feel it is important to address this issue briefly in the text and in more detail in the supplement. - Since the focus here is on MAGs and FCI, I think the example in Figure 1 should also correspond to a MAG. The current graph in figure 1 couldn't be learned by FCI. The alternative would have to be enumeration of all acyclic directed mixed graphs consistent with the learned partial ancestral graph using a method like [c] https://aaai.org/ocs/index.php/AAAI/AAAI17/paper/view/14951. - Rule 8 should reduce to running FCI with background knowledge that no edges can point into vertices in {\cal F} and as mentioned before, the additional knowledge that Fx -> X always exists should also be enforced. - The formulation of rule 9 is incorrect or imprecise. The definition of an inducing path is incorrectly used in rule 9, line 301: " we can conclude that every non-endpoint node along the inducing path is adjacent to Y in MAG(AugI(D))". The presence of an inducing path between Fx and Y does imply that Fx and Y will be adjacent in the corresponding MAG. However, it does not imply that every non-endpoint along the inducing path is adjacent to Y. The correct definition is used in the proof of soundness of rule 9 in the supplement. An inducing path between A and B is defined as a path from A to B where every non-endpoint is a collider and an ancestor of at least one of the endpoints A, B. In the PAG learned in Figure 2(c), this criterion is fulfilled by orienting either X o->Y to X->Y or orienting W o-> Y to W -> Y making X an ancestor through X -> W -> Y. Unless there are other properties of Fx (which don't seem to be immediately clear to me and if present should be made more explicit) that determine the orientation of the X o-> Y edge, it seems that X can be an ancestor of Y in multiple ways. While I see mention that the inducing path must pass through X, it is not clear why X could not be an ancestor in different ways. If rule 9 is invalid in this respect, this affects the soundness of the procedure. - The conclusion is great, I thank the authors for precisely pointing out what extensions and open problems may still be worked on.

[Author Response · NeurIPS 2019]

We would like to thank the reviewers for their comments and constructive feedback. We will implement all the minor
comments by the reviewers. Below, we address the main issues raised and clarify some misunderstandings.

**R2:** *Limitation of intervention not changing parent set:* There are many settings in the empirical sciences where
interventions do not change the parents. For instance, a drug injected into a biological system may change the baseline
of certain proteins (the underlying causal mechanisms) but it neither entirely suppresses their expression nor create new
connections. Also, the work of Yang et al. (2018) characterizes soft interventions in systems without latent variables.
Mooij et al. (2013) discussed interventions of this nature in the context of equilibrium in cyclic causal models.

*Usage of MAGs:* The reviewer's observation only holds for hard interventions. Within the soft intervention setting,
Augmented MAGs actually do not lose information with respect to the tests we consider. Fig. 2 is excellent to
illustrate this: Since the intervention is soft, the inducing path between $X$ and $Y$ through $Z$ is not eliminated upon $X$'s
intervention. Thus, we cannot rule out $X - Y$ adjacency. In general, soft interventions do *not* break inducing paths.

*Comparison to nested Markov models:* Nested Markov models consider constraints that are computable from the
observational distributions, also known as the Verma constraints, as pointed out by the reviewer. We, on the other hand,
focus on constraints imposed by a given set of interventional distributions that are not necessarily computable from
observational data. Moreover, we provide a graphical characterization to the proposed interventional equivalence class
while, to the best of our knowledge, there is no characterization for equivalence in nested Markov models.

*Can't we employ rule 9 directly on FCI output?:* Consider the diagram in Fig. 2a. By running FCI on observational
data, we would get the following PAG $\{X \circ\!\!\rightarrow Y, X \circ\!\!-\!\!\circ Z, X \circ\!\!-\!\!\circ W, Z \circ\!\!\rightarrow Y, W \circ\!\!\rightarrow Y\}$ . Our approach recovers more
orientations even without Rule 9 as shown in Fig. 2c. Note that F-nodes are not variables, but they are instrumental to
recover additional invariant graphical properties such as orientations through our newly proposed machinery.

*Maximality of Augmented MAG:* The augmented MAG is indeed the maximal ancestral graph constructed from Aug(D).
The current definition does not reflect this, as correctly observed by the reviewer and we will update it.

**R3:** *Controlled experiment assumption:* We apologize for the confusion. Your observation that the assumption is
required in the proof of Theorem 4 is correct. Without this assumption, the proposed characterization would still
apply under a different F-node construction: Instead of an F-node for every symmetric difference, we would use an
F-node for the variables whose mechanism differ across interventions. For instance, if $I = \{A, B, C\}$, $J = \{B, C\}$,
$P_I(B|pa_B) \neq P_J(B|pa_B)$, $P_I(C|pa_C) = P_J(C|pa_C)$, then $F_{I \triangle J} \rightarrow \{A, B\}$. We will clarify this with an example.

**R4:** *Faithfulness issues of [9]:* Note that similar to Yang et al. (2018), we do not treat F-nodes as random variables, but
simply devices that are instrumental in graphically representing the results of the invariance tests formalized in the
paper. Therefore, faithfulness involving F-nodes does not arise in our approach. Still, a different type of faithfulness
assumption is still required to map test results directly back to its graphical representation (Def. 7, pg. 7 lines 285-286).

*Advantages over [9]:* To the best of our knowledge, Theorem 2 in the paper is the first characterization of an
interventional equivalence class under causal insufficiency (presence of latents). As for the algorithms, the FCI-JCI
algorithm of [9] is indeed similar to ours, although the resultant graphs are different (they consider regime variables
to be random variables connected via bidirected edges). One obvious difference due to our characterization is Rule 9
which can orient additional edges, see Fig. 2 in pg. 8.

*Edges between F-nodes and non-intervened variables:* The adjacency between F-nodes and nodes outside the
corresponding intervention has an important role in our characterization. They signify the existence of an inducing path
between the intervention set and the node outside the intervention. For instance, the adjacency of $F_X$ and $Y$ in Fig. 2b
encodes the presence of an inducing path between $X$ and $Y$ through $Z$ in Fig. 2a. Meanwhile, we still cannot rule out
the presence of a directed edge from $X$ to $Y$ since such a causal diagram would entail the same invariance tests. These
edges in the augmented MAG allows us to capture all the test results using a single graph.

*Comments on Fig. 1:* Our goal with Fig. 1 is to illustrate that, for a ground truth causal diagram with latent variables
(Fig. 1a), and for a given set of interventions ($p_x$, $p$ in Fig. 1b, $p_x$, $p$, $p_{xz}$ in Fig. 1c), F-node separation statements can
be used to "store", or represent, the results of do-calculus tests. Our goal is not only to learn the MAG, but mainly to
establish a graphical equivalence relation that maps directly to the equivalence of the corresponding do-calculus tests.

*Soundness of Rule 9:* Consider an F-node $F_k$ and $Y \notin S_k$. If there is an inducing path $p$ between them, then every
observed non-endpoint $Z$ along the path must be a collider and an ancestor of either endpoint. The crucial point is that
$Z$ cannot be an ancestor of $F_k$ since F-nodes are root nodes by construction. Then, $Z$ must be an ancestor of $Y$. It
follows that the subpath of $p$ from $Z$ to $Y$ is an inducing path as well since every observed non-endpoint is a collider and
ancestor of $Y$, i.e., one of the endpoints. Thus, $Z$ and $Y$ are adjacent in the augmented MAG and the edge is directed
out of $Z$, since $Z$ is an ancestor of $Y$ as described. As for Fig. 2c, $X$ could be an ancestor of $Y$ through different ways
in the causal diagram as suggested, but this implies that the edge is out of $X$ by the ancestral property of MAGs.

[Meta-Review · NeurIPS 2019]

This paper proposes a new equivalence theory for MAGs learned from a combination of observational and (soft) interventional data, and propose a modification of the FCI algorithm that learns the resulting classes. The reviewers found the paper well-structured and a good contribution to the causal discovery literature.